biomechanics/psychology/human-computer interaction

virtual reality, interpersonal coordination, stepping behaviour, walking avatars, gait biomechanics

**Author for correspondence:**
Artur A. Soczawa-Stronczyk
e-mail: aamss1@leicester.ac.uk

# Gait coordination in overground walking with a virtual reality avatar

## Artur A. Soczawa-Stronczyk[1] and Mateusz Bocian[1,2]

[1]School of Engineering, and [2]Biomechanics and Immersive Technology Laboratory, University of Leicester, Leicester, UK

AAS-S, 0000-0002-3393-8404; MB, 0000-0002-3539-5474

Little information is currently available on interpersonal gait synchronization in overground walking. This is caused by difficulties in continuous gait monitoring over many steps while ensuring repeatability of experimental conditions. These challenges could be overcome by using immersive virtual reality (VR), assuming it offers ecological validity. To this end, this study provides some of the first evidence of gait coordination patterns for overground walking dyads in VR. Six subjects covered the total distance of 27 km while walking with a pacer. The pacer was either a real human subject or their anatomically and biomechanically representative VR avatar driven by an artificial intelligence algorithm. Side-by-side and front-to-back arrangements were tested without and with the instruction to synchronize steps. Little evidence of spontaneous gait coordination was found in both visual conditions, but persistent gait coordination patterns were found in the case of intentional synchronization. Front-to-back rather than side-by-side arrangement consistently yielded in the latter case higher mean synchronization strength index. Although the mean magnitude of synchronization strength index was overall comparable in both visual conditions when walking under the instruction to synchronize steps, quantitative and qualitative differences were found which might be associated with common limitations of VR solutions.

## 1. Introduction

Spontaneous gait coordination in walking has become one of the canonical examples of an emergent behaviour between autonomous agents [1–3]. Persistent gait coordination patterns have been reported in dyadic walking in side-by-side [4–13] and front-to-back [14,15] pedestrians' arrangements. The gait coordination strength and directionality has been shown to strongly depend in this case on the type and the amount of available sensory information. However, the results from dyadic walking cannot be simply extrapolated to walking in groups and

crowds—a situation commonly encountered in real-world settings, while data enabling gait coordination patterns to be examined in the latter case are still sparse [16–19].

Only a few studies investigated spontaneous gait synchronization in real-life environment [4,16,17,19]. The main reasons for this pertain to controllability of experimental conditions and observability of measured variables. On the one hand, the desire of closely controlled setting is difficult to realize in an environment subjected to various disturbances. On the other hand, the inference of spatial and temporal gait variables demands a distributed instrumentation system capable of simultaneous capture of data generated by multiple pedestrians. Consequently, alternative methods of investigating gait adaptations between pedestrians were proposed, predominantly relying on treadmills. However, the ecological validity of results from studies other than those enabling overground walking can be put in question [20,21].

Considering these limitations, virtual reality (VR) technology is thought to have the potential to boost the efficiency of research on locomotion. The current state of VR technology allows users to interact with near-realistic virtual environments, including autonomous agents. To this day, VR locomotion studies have been concerned with spatial cognition and awareness [22–24], locomotor training [25–33], locomotion and perception in real-world and virtual environment [34–39], navigation [40–42], collision avoidance [43,44] and crowd behaviour [45,46]. Locomotion interfaces allowing users to interact with and explore the VR content include a joystick [47], walking-in-place [48,49], treadmill [50–56] and those enabling overground walking compatible with real-world experience [35,36,57]. The last solution shows the highest promise in capturing pedestrians' natural behaviour [40–42,49,58,59], assuming it employs isometric rather than non-isometric mapping. The former technique retains motion parameters such as translation and rotation, and their time derivatives, of those from the real environment in the virtualized equivalent, whereas the latter modifies some or all of these parameters [60]. (For a detailed review of developments on human locomotion in VR the reader is referred to Steinicke *et al.* [38].)

However, studies on the overground locomotion in VR, employing isometric mapping, are still rare and their main focus remains the gait kinematics and sensory integration rather than interaction with VR-based agents [35,57,61]. Furthermore, the gait enabled in these studies by VR interfaces can hardly be considered unconstrained, as the walkers are either accompanied by a man-handled trolley carrying PC provision or their locomotion is realized over few gait cycles.

It is important to note that VR comes with certain inherent limitations. Previous comparative studies on gait parameters during walking in real and virtual environment found underestimation of egocentric distances [62] as well as decrease of the walking speed and the stride length a common response to VR immersion [63]. This was suggested to be caused by the altered distance perception in virtual reality, often referred to as distance compression [36,64–67], which may become negligible after a 5 min period of habituation to virtual environment [68]. Some more recent studies found walking and the motion adaptation in VR to be quantitatively comparable to those in the real-world environment [36,43,44,69,70]. A widely acknowledged consensus is that while quantitative differences between virtual and real-life locomotion might still exist, the pedestrian behaviour in both environments is qualitatively compatible [34,43,44,71,72].

To address the gap in current knowledge, the ambition of this study is to develop a VR platform enabling unconstrained overground walking in the presence of virtual agents while preserving ecological validity with regards to the behaviour in real-life settings. The main advantages of this solution in human locomotion studies derive from controllability and observability [44,46], typically associated with testing in a laboratory environment, and the ability of testing complex, uncommon or even unrealistic scenarios [73]. To validate the platform, six subjects were asked to walk with a real and virtual pacer in various topological arrangements while having their gait monitored. Gait kinematics were analysed to identify any coordination patterns.

It was hypothesized that a VR locomotor interface enabling unconstrained overground walking can evoke gait coordination patterns between a real pedestrian and a virtual reality avatar similar to those observed when walking in the corresponding conditions in the real-life environment. Informed by the previous studies on walking in pairs [9,10] and groups [16–19], it was expected that the spontaneous synchronization of gait between walkers, if present, would be of transient nature [9,10]. Given a sufficiently long walking path is provided, it was expected that the spontaneous synchronization of gait would be relatively weak. Due to a lesser number of available synchronization stimuli when compared to walking in a group of pedestrians [16,19], it was expected that the synchronization strength in dyadic walking would be, on average, higher than that for walking in a group. Furthermore, it was expected that the instruction to synchronize steps would cause the synchronization strength to be significantly

higher than in the case of lack thereof [19]. Finally, it was expected that the synchronization strength for walking front-to-back would be higher than that for walking side-by-side [16,19].

The rest of the paper is organized as follows. Section 2 introduces the developed VR platform, test subjects, experimental protocol, instrumentation and data processing. The numerical results are provided in §3. Section 4 presents the discussion of the results in the context of previous studies. Conclusions are given in §5.

# 2. Method

## 2.1. Virtual reality platform

A bespoke experimental platform was developed for the purpose of this study, relying on an immersive VR environment and a distributed motion capture system. The VR environment contained a virtual pedestrian, of which gait characteristics could be closely controlled, within a space closely resembling a real-world environment in which the experimental campaign took place. The main idea underlying the development of the experimental platform was to enable kinematic gait data to be recorded during dyadic walking with a pacer, that being either a real human or their anatomically and biomechanically representative VR avatar.

The following three main steps were implemented to create a VR platform enabling overground walking in the presence of an avatar. Firstly, walking cycles of a real person were recorded using a motion capture suit (MCS) to serve as a blueprint for the avatar's movement. The recordings were taken with a real person walking along straights and turns at various pacing frequencies. Secondly, a realistic humanoid character to be presented in VR had to be created together with an animation controller responsible for driving the movement of the avatar. Thirdly, a state-of-the-art navigation system, using artificial intelligence, was set up and used to move the avatar within the VR environment.

### 2.1.1. Motion capture

Eight OptiTrack Prime 13 cameras were used to record the motion of a 25-year-old male performer (height: 1.825 m, weight: 80 kg) wearing a motion capture suit equipped with 37 reflective markers placed on body landmarks. All data were recorded at the sampling rate of 120 Hz and transferred through a gigabit Ethernet network compliant with the IEEE 802.3 standard to the processing unit, where they were logged and post-processed using proprietary Motive:Body 2.1.1 software [74]. Four types of walks were performed at pacing frequencies, $f_p$, ranging from 1.3 to 2.0 Hz, at 0.1 Hz increments, enforced with an audible beat of a metronome: walk in a straight line and along circular arcs with the radii of 0.635 m (25 in), 1.27 m (50 in) and 2.54 m (100 in).

The initial MCS data post-processing was done in Motive:Body software, where gaps in markers' positions were filled using cubic or linear polynomial interpolants, depending on the trajectory of a marker. The raw data were smoothed by applying a low-pass fourth-order two-way Butterworth filter with the cut-off frequency of 6 Hz. This was done to remove any motion artefacts caused by the MCS's fabric movement and any changes in the tracking quality due to fluctuations in the lighting conditions and ambient temperature. The final stage of post-processing was performed in Autodesk MotionBuilder 2018 [75], where the MCS data were down-sampled to 30 frames s$^{-1}$. The software enabled to adjust details, such as feet's floor contact and fingers' positions. Finally, to minimize the repeatability of performer's motions and to preserve the variability of gait, multiple gait cycles recorded at the same pacing frequency and walking path during different takes were extracted. They were then stitched together in an arbitrary order to create looping sequences of performer's motion containing an integer number of gait cycles, each slightly different than another.

### 2.1.2. Avatar creation

A virtual representation of the performer to be incorporated within VR was created in Adobe Fuse CC 1.2 [76]. The geometry and appearance of body segments, including face expression, were chosen to match, as closely as possible, those of the performer. The character, hereafter referred to as the avatar, was then imported into (Adobe) Mixamo web service [77] where it was rigged, i.e. where its skeleton was created and tied with its geometry to facilitate the control of avatar's motion. The avatar was then imported into Unity 2018.4.0f1 game engine [78] where a custom animation controller was built using pre-recorded performer's walking sequences, as outlined in §2.1.1. The animation controller relied on a

**Table 1.** Basic information about the test subjects of the experiment.

| | test subject ID | | | | | |
|---|---|---|---|---|---|---|
| | S1 | S2 | S3 | S4 | S5 | S6 |
| gender | female | male | male | male | male | male |
| age | 28 | 30 | 36 | 47 | 22 | 35 |
| mass (kg) | 77.0 | 80.1 | 81.6 | 74.4 | 84.9 | 87.8 |
| height (cm) | 172.0 | 191.6 | 173 | 189.5 | 180.5 | 183.5 |

linear blending algorithm based on two variables: the pacing frequency and the instantaneous direction of progression. Therefore, having been prompted by a command from within a custom-written Unity3D program, the avatar could walk in a manner closely resembling the performer in terms of anatomy, biomechanics and appearance at any pacing frequency (hence speed) and in any direction, in near real time.

The auditory stepping cues were incorporated by adding a sound source to each of avatar's feet. The sound sources were populated with a database of step sounds recorded on various walking surfaces, with 10 slightly different step sounds per surface. This allowed to play a non-repeating step sound, appropriate to the walking surface, every time the game engine detected a collision of avatar's foot with the virtual walking surface.

### 2.1.3. Steering system

Polarith AI 1.6 [79] was implemented to navigate the avatar around the VR scene. Polarith AI is a state-of-the-art fully programmable artificial-intelligence navigation system based on a multi-objective optimization algorithm. The workflow of the algorithm is divided into two stages. Firstly, it samples the scene to detect the position of obstacles and destination relative to the driven avatar. Secondly, it uses the inbuilt optimization algorithm to find the best local solution for the optimization problem [80]. The solution represents an optimal direction of movement which is fed directly to the avatar's animation controller, outlined in §2.1.2. A simple 'path follow' action was programmed for the purpose of the tests, where the avatar was following a pre-programmed path.

## 2.2. Test subjects

The basic information about six test subjects recruited for the study is given in table 1. Before taking part in the experiment, test subjects were required to familiarize themselves with an information letter, fill in a physical activity readiness questionnaire, and sign an informed consent form. Test subjects wore casual clothing and flat sole shoes.

## 2.3. Location

The Charles Wilson Sports Hall located within the main campus of the University of Leicester was chosen as an experimental location. The sports hall is 16.7 m wide and 33.5 m long and has the clear height of the ceiling at 5.6 m. The wooden floor of the hall was covered with a dark monotone carpet tiles for the purpose of the experiment, to mask the floor markings hence eliminating visual reference cues which could have affected test subjects' behaviour.

## 2.4. Experimental protocol

Test subjects were given habituation time to familiarize themselves with the experimental environment. Each test subject was then asked to perform 16 walks, each consisting of eight laps comprising two 10 m straights and two turns along half of the circumference of a circle with 5 m in diameter. During each walk, the test subject was asked to walk together with a pacer. The pacer was either the same male subject used to animate the avatar, as described in §2.1.1, hereafter referred to as Pacer[RL], or an animated virtual avatar, as described in §2.1.2, hereafter referred to as Pacer[VR]. Therefore, the movement of the pacer accompanying the test subject was compatible in the real-life (RL) and virtual environment (VR).

To account for any potential directional bias in pedestrian behaviour, each walk was performed in clockwise and anticlockwise directions along the looping path. Test subjects walked either next to the pacer, i.e. side-by-side on the inward of the loop, hereafter denoted SbS, or behind the pacer, i.e. front-to-back, hereafter denoted FtB. During the first eight walks, hereafter referred to as uninstructed synchronization and denoted US, the test subjects were only told to maintain their prescribed position relative to the pacer throughout the test. For the final eight walks, hereafter referred to as instructed synchronization and denoted IS, the test subjects were also instructed to walk in step with the pacer, such as to match the timing of their footsteps made with ipsilateral legs.

The pacing frequency of Pacer$^{RL}$ was controlled by using a pair of over–ears Pioneer SE-M521 headphones connected to KORG MA-1 metronome, whereas the pacing frequency of Pacer$^{VR}$ was directly set in the animation controller described in §2.1.2 The pacing frequency was established for each test subject individually, based on the Froude number, *FR*, equal to 0.15 and described by the following formula

$$FR = \frac{v}{gl},$$

where $v$ is the walking speed (expressed in m s$^{-1}$), $g$ is the gravitational acceleration (expressed in m s$^{-2}$) and $l$ is the test subject's leg length (expressed in m). The Froude number was used to obtain dynamically compatible gait patterns for all test subjects by compensating for differences in their anatomical (i.e. geometric) characteristics [81]. Due to practical reasons, the leg length was estimated based on the test subject's height and gender using the relationships established by Pheasant [82] and explicitly given in Bocian *et al.* [83]

$$l = \begin{cases} 0.7028\ h_m - 0.3091 \\ 0.6797\ h_f - 0.2781 \end{cases},$$

where $h_m$ and $h_f$ is the height for male and female test subjects (expressed in m), respectively.

Finally, the walking speed was converted to the corresponding pacing frequency, $f_p$, using the relationship obtained in Soczawa-Stronczyk *et al.* [19]

$$f_p = 0.66v + 0.99.$$

Following this procedure resulted in test subjects' walking with the mean speed (given with ± s.d. where applicable) of $v^{RL} = 1.28 \pm 0.07$ m s$^{-1}$ and $v^{VR} = 1.20 \pm 0.03$ m s$^{-1}$ in the RL and VR environments, respectively.

## 2.5. Instrumentation

A motion capture system consisting of 24 OptiTrack Prime 17W and 10 OptiTrack Prime 13 cameras was deployed in the sports hall. During VR tests, test subjects wore Oculus Rift CV1 head-mounted display (HMD) equipped with reflective markers to track its movement. The HMD was connected to MSI VR One 7RE backpack PC used to generate VR environment. HMD's positional data were streamed from Motive: Body 2.1.1 software running on MCS data processing unit to the backpack PC through a wireless network compliant with IEEE 802.11n-2009 [84] standard, using the NatNet 3.0.1 [85] server broadcast protocol. This was to ensure untethered experience, i.e. for the test subjects to move freely within the capture volume of the MCS. The MCS latency, defined as the time elapsed from the cameras' exposure to the tracking data packages fully solved by Motive software and ready for transmission over IEEE 802.11n-2009 wireless network, was measured *ex post facto*. It did not exceed 4.7 ms. The latency of the data transmission over IEEE 802.11n-2009 wireless network was estimated to be approximately 3 ms, based on the median value of a typical consumer grade, uncongested wireless network [86]. Although the latencies due to scene rendering and display in the HMD were not directly measured, a similar set-up was previously used in [43]. No discomfort due to the visual information delay was reported in *post hoc* interviews with the test subjects. If the latency had a strong debilitating effect on our results, we would have expected the mean phase difference to differ significantly between corresponding tests in RL and VR environment under the instruction to synchronize steps (IS). However, this was not the case, as discussed in §3.3.3.

The VR environment used in the tests, shown in figure 1, was created using ARCHICAD 23 software [87]—a state-of-the-art building information modelling (BIM) tool. It consisted of a highly detailed representation of the Charles Wilson Sports Hall including three-dimensional doors, windows, lighting features and basketball infrastructure. The walls and the ceiling were covered in solid colours and the floor was rendered using a dark monotone carpet tile texture with the resolution of 288 px m$^{-1}$.

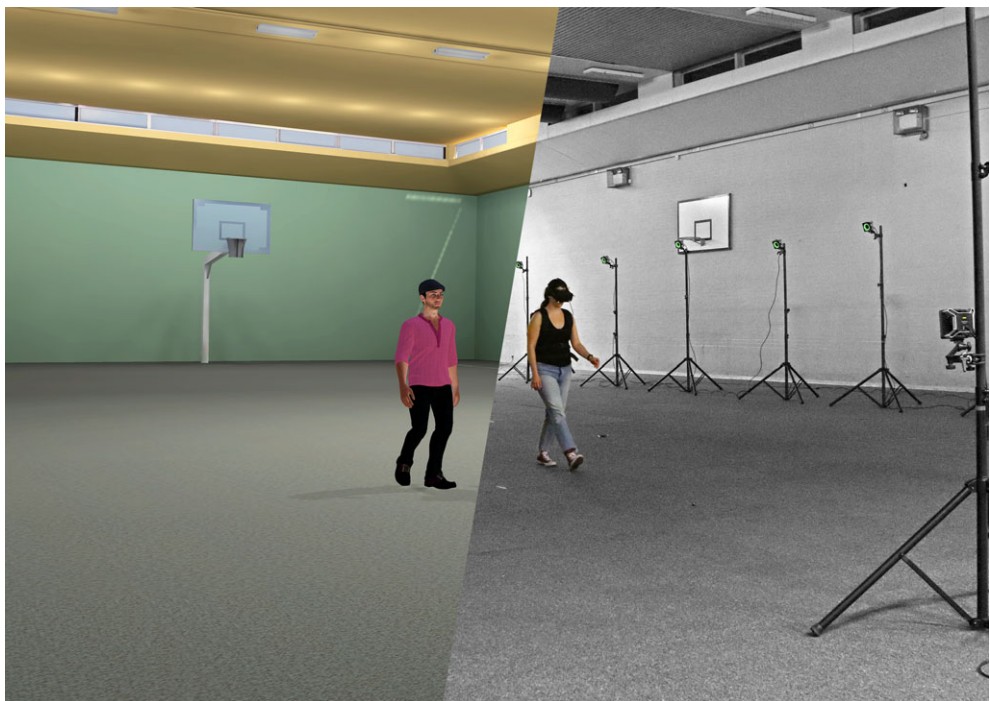

**Figure 1.** A collage with, on the right panel, test subject S1 wearing head-mounted display being tracked with a set of motion capture cameras and, on the left panel, a screenshot from virtual reality containing Pacer$^{VR}$.

The appearance of these components was closely matching the real-world environment. To increase realism, a global (Sun) light was positioned outside the modelled room. The global light was casting inner shadows, thus providing variable brightness surface patterns facilitating optic flow and distance estimation. An exemplar VR scene used during the experimental campaign is available as a part of the dataset supporting this study (see Data accessibility).

For all the walks, test subjects were instrumented with two APDM Opal™ wireless attitude and heading reference systems (AHRS), one attached at the level of fifth lumbar vertebra and the other one on the right ankle using elastic straps. The data recorded by the AHRS were time-locked and sampled at 128 Hz. Only the data from the ankle sensor were used in subsequent analysis, as they carried information allowing unequivocal determination of the stage of a gait cycle. For this purpose, the vertical component of the acceleration vector (i.e. that aligned with gravity) was extracted by resolving the recorded three-dimensional acceleration signals from the local (i.e. sensor) to the global coordinate system using quaternion algebra.

The Pacer$^{RL}$ was equipped with a set of AHRS of the same type and placed at the same body landmarks, and time-locked to those of the test subject thus enabling synchronization to be readily quantified based on compatible signals. However, this was impossible to realize during VR walks. In this case, the Pacer$^{VR}$ limb's displacement measured at the location corresponding to the placement of AHRS was recorded within the game engine at a sampling rate of approximately 50 Hz. Since that signal was not time-locked with the signals from AHRS, the following time alignment procedure was implemented. One AHRS was strapped to a rigid body tracked by the MCS and controlling the motion of an object in VR. The displacement of that virtual object was recorded by the game engine together with the displacement of the skeleton driving Pacer$^{VR}$. Before and after each of the VR walks, the rigid body with the attached AHRS was waved slowly in sinusoidal motion to create a signal for the time alignment of data from the two systems (i.e. AHRS and VR) in post processing.

## 2.6. Data analysis

All analyses of synchronization are based on vertical velocity signals from the ankle. This is because the acceleration signals from AHRS and displacement signals from Pacer$^{VR}$ had to be reconciled to a common physical quantity before further processing.

### 2.6.1. Signal processing

Pacer$^{VR}$'s signal was up-sampled from the sampling rate of approximately 50 to 128 Hz using *resample* function [88] implemented in Matlab R2019b [89], to match the sampling rate of the signals from AHRS monitors. Next, the AHRS signals (acceleration in m s$^{-2}$) and the up-sampled signals from the game engine (displacement in m) were brought to a common kinematic variable, that being velocity (expressed in m s$^{-1}$). To numerically integrate the AHRS' acceleration signal, a fourth-order two-way band-pass Butterworth filter was used with the frequency band set to preserve the first three harmonics of the original signal. The signal was integrated using the cumulative trapezoidal numerical integration method [90] and then filtered using high-pass, fourth-order Butterworth filter with the cut-off frequency equal to half of the frequency of the first harmonic. The numerical differentiation of the game engine's displacement signal was performed by calculating a one-dimensional numerical gradient of the displacement vector aligned with gravity and dividing it by the constant numerical gradient of the corresponding time. The resulting velocity vector was filtered using the fourth-order band-pass Butterworth filter with the frequency band set to preserve the first three harmonics of the signal, as in the case of numerical integration.

Having obtained signals representing the same physical quantity, it was possible to use two signals from the rigid body–AHRS couple to find the delay using *finddelay* function [88] in Matlab R2019b and align them in time.

The exemplar results of the signal processing procedures described in this section are shown in figure 2*a*,*b*. Figure 2*c*,*d* is included here for the clarity of presentation, although the relevant discussion is given in §2.6.2.

Two types of signals were used in the subsequent analysis: (i) the original signals, containing the entire walking path, hereafter referred to as loops, and (ii) the truncated signals, created by removing sections of the loops containing data from the turns, hereafter referred to as straights. The locations of the individual straights in the velocity signal were identified by using the angular velocity data recorded by the AHRS attached at the test subject's ankle.

For both types of signal, the first and the last straight sections of each walk were discarded, as they contained gait initiation and termination stages, respectively. This resulted in 15 straights per each walk type, with the exception being the virtual anticlockwise FtB walk under US experimental conditions undertaken by test subject S5 who performed seven laps around the path instead of eight.

### 2.6.2. Quantification of synchronization

Following previous studies [8–10,12,13], an analytic signal was employed herein to derive an instantaneous phase of the captured signals. Prior to obtaining the analytic signal, the velocity signals from the ankle were band-pass filtered using the fourth-order Butterworth filter with a frequency band limited by 0.70 times the minimum stride frequency and 1.25 times the maximum stride frequency from the pair of considered walkers, as was the case in van Ulzen *et al.* [9]. The analytic signal, containing information on the instantaneous amplitude and phase angle, was obtained using Hilbert transform, commonly defined as [91]

$$v^a(t) = v(t) + \frac{i}{\pi} P.V. \int_{-\infty}^{+\infty} \frac{v(t')}{t - t'} \, dt',$$

where $t$ is the real-valued time variable, $v(t)$ is the real-valued velocity signal, $v^a(t)$ is its analytic representation and $P.V.$ is the Cauchy principal value of the integral. The instantaneous phase angle time series, $\phi(t)$, was obtained by calculating the angle between the real, $\Re$, and imaginary, $\Im$, parts of the analytic signal [92]

$$\phi(t) = \tan^{-1} \frac{\Im[v^a(t)]}{\Re[v^a(t)]}.$$

The bivariate phase difference signal, $\phi_{p,s}(t)$, was obtained by subtracting test subject's phase angle time series, $\phi_s(t)$, from pacer's phase angle time series, $\phi_p(t)$

$$\phi_{p,s}(t) = \phi_p(t) - \phi_s(t).$$

An exemplar signal representing the evolution of phase difference in time is shown in figure 2*c*, and the corresponding histogram of phase difference is shown in figure 2*d*. The rate of change of phase difference varies throughout the record, such that it is the fastest when passing through the extreme values and slows down around the state indicating no phase difference. Consequently, the phase difference distribution is skewed towards the values at and around 0 rad.

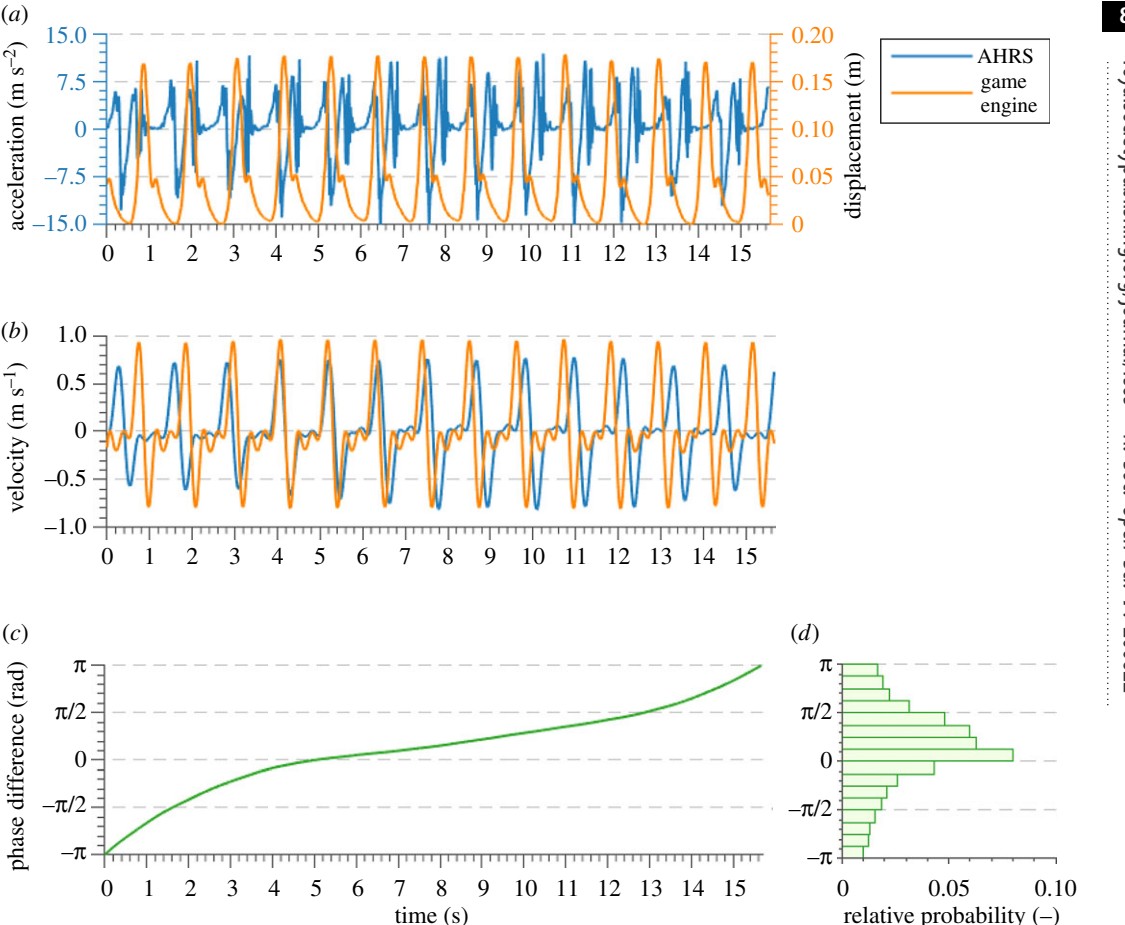

**Figure 2.** Preview of (*a*) AHRS acceleration and game engine displacement signals from the ankle, with (*b*) the results of the numerical integration and differentiation of the acceleration and displacement signals, respectively, as well as (*c*) the time series of phase difference and its (*d*) relative probability distribution. All presented signals were recorded during anticlockwise side-by-side walk with VR avatar under US experimental condition for test subject S2.

The bivariate synchronization strength index, $\rho_{p,s}$, was then calculated, based on the Shannon entropy, $E_{p,s}$, of the phase difference distribution [92]

$$E_{p,s} = -\sum_{k=1}^{N} P_{p,s}^{k} \ln P_{p,s}^{k},$$

where $P_{p,s}^{k}$ is the probability of the phase difference, $\phi_{p,s}(t)$, falling into a $k$th bin of $\pi/8$ rad in size and $N$ is the total number of bins. To allow for synchronization strength index to be comparable for different walks and test subjects, the index was normalized by the maximum attainable Shannon entropy in the case of the perfect phase synchronization

$$\rho_{p,s} = \frac{\ln N - E_{p,s}}{\ln N}.$$

The index can take values from 0 to 1, where 0 relates to a uniform distribution of phase difference, i.e. lack of synchronization, whereas 1 corresponds to Dirac-like distribution of phase difference, i.e. perfect synchronization.

## 3. Results

### 3.1. Average gait characteristics

The average stride frequency, $\bar{f}$, and average stride length, $\bar{l}$, were calculated for each test to evaluate the difference in gait characteristics between walking in RL and VR environment. The Welch's $t$-test [93] at

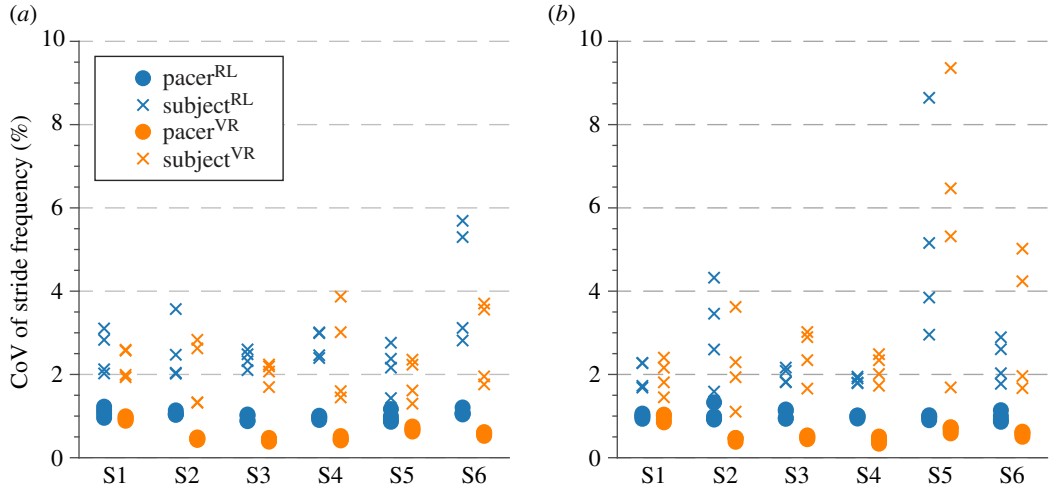

**Figure 3.** The coefficient of variation of the stride frequency of pacers and test subjects obtained (*a*) without and (*b*) with the instruction to synchronize gait.

5% significance level ($p = 0.05$) was then used to test the hypotheses that the means of $f$ and $l$ were equal. The data collected under US only were used in this analysis since the instruction to synchronize steps had overridden naturally occurring gait patterns, as discussed in §4.3.1 Moreover, walks in SbS and FtB were considered separately due to the difference in the distance travelled by test subjects in these two topological arrangements.

The average stride frequency, obtained using the fast Fourier transform, during walking SbS was $\bar{f}_{SbS}^{RL.US} = 0.829 \pm 0.043\,Hz$ and $\bar{f}_{SbS}^{VR.US} = 0.822 \pm 0.053\,Hz$ in the RL and VR environment, respectively. The corresponding values for walking FtB were $\bar{f}_{FtB}^{RL.US} = 0.832 \pm 0.032\,Hz$ and $\bar{f}_{FtB}^{VR.US} = 0.852 \pm 0.044\,Hz$, respectively. Hence the test subjects' stride frequency in VR environment was on average lower by 0.77% and higher by 2.35% for walking SbS and FtB, respectively. In both cases, the difference was statistically insignificant, $p_{SbS} = 0.748$ and $p_{FtB} = 0.226$.

The average stride length for walking SbS was $\bar{l}_{SbS}^{RL.US} = 1.38 \pm 0.10\,m$ and $\bar{l}_{SbS}^{VR.US} = 1.26 \pm 0.09\,m$ in the RL and VR environment, respectively. The corresponding values for walking FtB were $\bar{l}_{FtB}^{RL.US} = 1.42 \pm 0.08\,m$ and $\bar{l}_{FtB}^{VR.US} = 1.41 \pm 0.09\,m$, respectively. Hence the test subject's stride length in VR environment was on average lower by 8.19% and 2.00% for walking SbS and FtB, respectively. The difference was statistically significant for walking SbS, $p_{SbS} = 0.010$, and statistically insignificant for walking FtB, $p_{FtB} = 0.402$.

In a similar fashion, pacers' average stride frequencies were extracted and compared to those of test subjects for corresponding walks. Under US experimental condition, the difference was equal to $\bar{f}_{P-S}^{RL.US} = 0.058 \pm 0.044\,Hz$ and to $\bar{f}_{P-S}^{VR.US} = 0.052 \pm 0.054$ in the RL and VR environment, respectively. Hence, on average, the pacer walked with a higher stride frequency than the test subject.

A closer match between stride frequencies was found for IS experimental condition. The average stride frequency difference was $\bar{f}_{P-S}^{RL.IS} = 0.005 \pm 0.017\,Hz$ and $\bar{f}_{P-S}^{VR.IS} = 0.006 \pm 0.017$ in RL and VR environment, respectively. Hence, on average, the pacer walked with a slightly higher stride frequency than the test subject.

## 3.2. Gait cycle variability

One of the requirements of the experimental protocol was to retain similar gait variability of the pacer in all experimental conditions in order to preserve this natural gait characteristic and obtain a compatible set of results. To verify this condition, the coefficient of variation of the stride frequency, $f_{CoV}$, was calculated to assess the gait variability of the pacer and all test subjects. The outcomes are shown in figure 3. Overall, the timing of the pacer's footsteps was repetitive, which is represented by the CoV values not exceeding 1.33% during any of the walks. It is worth pointing out that, due to the way the experimental campaign was designed, the pacer's gait variability was not directly affected by the instruction to synchronize, i.e. the change of the experimental conditions from US to IS. This is to say that the effort spent to coordinate the timing of footsteps was unidirectional, with the test subjects adjusting their steps to match those of the pacer. Moreover, the Pacer$^{VR}$ was hard-programmed to a pre-established average pacing

frequency while accounting for certain variability in the timing of footsteps (see §2.1), and by its very nature remained unaffected by test subject's behaviour throughout the duration of the test. However, even though subtle, the gait variability difference between the human and AI-driven pacer was distinguishable. The mean CoV of the stride frequency for Pacer$^{RL}$ is $\bar{f}_{CoV.P}^{RL} = 1.02 \pm 0.10\%$, with the minimum and maximum values of min $f_{CoV.P}^{RL} = 0.86\%$ and max $f_{CoV.P}^{RL} = 1.33\%$. The corresponding mean value for Pacer$^{VR}$ is $\bar{f}_{CoV.P}^{VR} = 0.59 \pm 0.19\%$, with the minimum and maximum values of min $f_{CoV.P}^{VR} = 0.34\%$ and max $f_{CoV.P}^{VR} = 1.03\%$.

Overall, the gait variability of test subjects was higher than that of the pacer; however, for most of the walks the CoV of test subjects' stride frequency did not exceed 6%. Under the US, the mean CoV of test subjects' stride frequency was $\bar{f}_{CoV.S}^{RL.US} = 2.76 \pm 0.96\%$ for walks in the RL environment and $\bar{f}_{CoV.S}^{VR.US} = 2.24 \pm 0.74\%$ for walks in the VR environment. Under the IS, the mean CoV of test subjects' stride frequency was $\bar{f}_{CoV.S}^{RL.IS} = 2.72 \pm 1.56\%$ for walks in the RL environment and $\bar{f}_{CoV.S}^{VR.IS} = 2.96 \pm 1.91\%$ for walks in the VR environment.

The highest gait variability under US was found for S6 in RL, at max $f_{CoV.S6}^{RL.US} = 5.69\%$, and for S4 in VR, at max $f_{CoV.S4}^{VR.US} = 3.87\%$. Overall, the gait variability under US did not show any systematic relationships between test subjects. The highest gait variability under IS was consistently found for S5, with max $f_{CoV.S5}^{RL.IS} = 8.64\%$ and max $f_{CoV.S5}^{VR.IS} = 9.36\%$ for RL and VR walks, respectively.

## 3.3. Synchronization strength

### 3.3.1. Statistical analysis

A Wilcoxon signed-rank test [94] was performed using IBM SPSS Statistics 26 [95] to determine the effect of the type of the analysed signal (loops or straights) on the bivariate synchronization strength index. The examination of the difference scores revealed near symmetrical data distribution. The difference between the two types of the signal was not statistically significant, $z = -1.07$, $p = 0.286$. As a consequence, only the values for the truncated signals, consisting of straights, will be presented throughout this paper, to ensure a level of compatibility between the outcomes of this and previous studies carried out along straight paths.

To gain an insight into general dependencies between the bivariate synchronization strength index and the experimental conditions, the first-order multiple linear regression analysis [96] was therefore performed adopting the significance level of 5% ($p = 0.05$). The statistical model took the following factors as the explanatory variables: (i) the environment in which the test took place (real-life environment, RL, or virtual environment, VR); (ii) the presence of the instruction to synchronize (uninstructed synchronization, US, or instructed synchronization, IS); (iii) the walking direction (clockwise or anticlockwise); and (iv) the relative position of the walkers (side-by-side, SbS, or front-to-back, FtB). The results showed that the presence of the instruction to synchronize had the strongest influence on the synchronization strength index, $t = 17.92$, $p < 0.0005$. This was followed by a much weaker, yet statistically significant influence of the environment, $t = -2.04$, $p = 0.045$. The remaining explanatory variables were not statistically significant predictors of the synchronization strength index. Therefore, from here onwards, the data will be presented separately for each of the environmental and experimental conditions and the walkers' collocation. No distinction will be made between clockwise and anticlockwise walking direction.

The synchronization strength indices are presented in figure 4a,b and accompanied by the mean circular direction indicating the directionality of the phase difference distribution in figure 4c,d. The mean circular direction, $\bar{\mu}$, was calculated by transforming all phase difference values into a two-dimensional vector $\boldsymbol{\mu} = (\cos \alpha, \sin \beta)$ and averaging over the number of data points [97]. According to the adopted convention, negative and positive values indicate the test subject is lagging and leading the pacer, respectively.

### 3.3.2. Uninstructed synchronization

The mean values of the synchronization strength index obtained for US in the two visual environments are rather low. In the RL environment, these values are $\bar{\rho}_{SbS}^{RL.US} = 0.063 \pm 0.097$ and $\bar{\rho}_{FtB}^{RL.US} = 0.060 \pm 0.110$ for walking SbS and FtB, respectively. In the VR environment, the corresponding values are $\bar{\rho}_{SbS}^{VR.US} = 0.067 \pm 0.122$ and $\bar{\rho}_{FtB}^{VR.US} = 0.016 \pm 0.018$. Only three synchronization strength indices reach above 0.21, previously suggested to signify the synchronization threshold for walking in pairs [12].

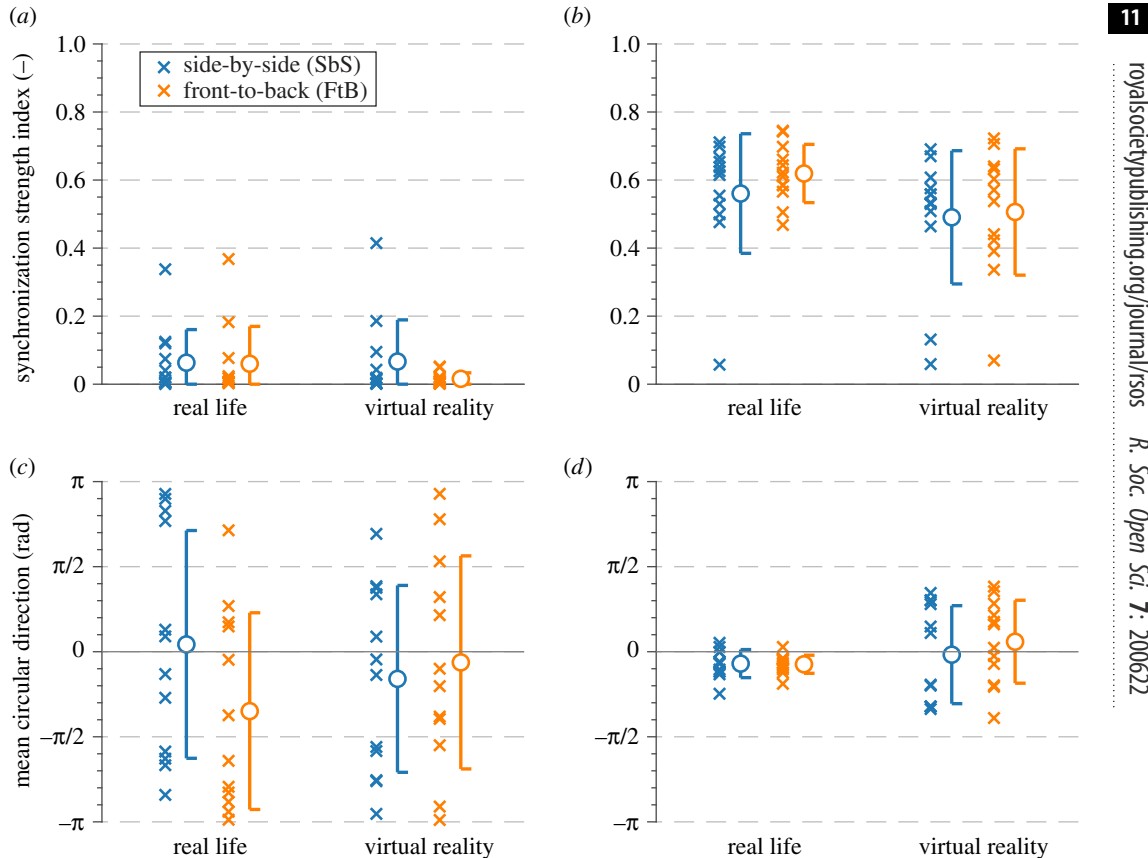

**Figure 4.** The results in terms of (a,b) synchronization strength indices and (c,d) mean circular direction values obtained (a,c) without and (b,d) with the instruction to synchronize gait. The values are presented with their respective means (denoted by circles) and ± s.d. intervals (denoted by whiskers).

The directionality of synchronization, quantified by the mean circular direction, is $\bar{\mu}_{\mathrm{SbS}}^{\mathrm{RL.US}} = 0.04\pi \pm 0.67\pi$ rad and $\bar{\mu}_{\mathrm{FtB}}^{\mathrm{RL.US}} = -0.35\pi \pm 0.58\pi$ rad for walking SbS and FtB in RL, respectively. In the VR, the corresponding values are $\bar{\mu}_{\mathrm{SbS}}^{\mathrm{VR.US}} = -0.16\pi \pm 0.55\pi$ rad and $\bar{\mu}_{\mathrm{FtB}}^{\mathrm{VR.US}} = -0.06\pi \pm 0.63\pi$ rad. Considering the results for both visual environments, the negative and positive values of the mean circular direction were recorded in 58.3% and 41.7% of cases, respectively.

### 3.3.3. Instructed synchronization

The mean values of the synchronization strength index for all walks obtained for IS exceed 0.21, previously suggested to signify the synchronization threshold for walking in pairs [12]. In the real environment, these values are $\bar{\rho}_{\mathrm{SbS}}^{\mathrm{RL.IS}} = 0.560 \pm 0.176$ and $\bar{\rho}_{\mathrm{FtB}}^{\mathrm{RL.IS}} = 0.619 \pm 0.086$ for walking SbS and FtB, respectively. In the virtual environment, the corresponding values are $\bar{\rho}_{\mathrm{SbS}}^{\mathrm{VR.IS}} = 0.490 \pm 0.196$ and $\bar{\rho}_{\mathrm{FtB}}^{\mathrm{VR.IS}} = 0.506 \pm 0.186$. The synchronization strength index values obtained for walking in RL are, on average, higher by 15.6% than their VR counterparts. Considering the results for both visual environments, the highest mean values of the synchronization strength index were obtained for walking FtB.

In the real environment, the mean values of the mean circular direction are $\bar{\mu}_{\mathrm{SbS}}^{\mathrm{RL.IS}} = -0.07\pi \pm 0.08\pi$ rad and $\bar{\mu}_{\mathrm{FtB}}^{\mathrm{RL.IS}} = -0.07\pi \pm 0.05\pi$ rad for walking SbS and FtB, respectively, with 91.7% of data points taking negative values. In the virtual environment, the corresponding values are $\bar{\mu}_{\mathrm{SbS}}^{\mathrm{VR.IS}} = -0.02\pi \pm 0.29\pi$ rad and $\bar{\mu}_{\mathrm{FtB}}^{\mathrm{VR.IS}} = 0.06\pi \pm 0.24\pi$ rad, with only 45.8% of data points taking negative values.

## 4. Discussion

### 4.1. Average gait characteristics

The average stride frequency and stride length of test subjects for walking SbS and FtB under US was similar between the corresponding tests conducted in RL and VR environment. This is apart from the average stride

length in walking SbS for which there was a statistically significant difference. However, this occurrence cannot be considered as evidence supporting the notion of gait parameters being adversely affected by the VR immersion. This is because the average duration of tests in RL and VR environment was in this case different, at 206 and 230 s, respectively. This is not the case for walking FtB, for which the average duration of tests in RL and VR environment was similar, at 231 and 228 s, respectively. In both cases (SbS and FtB walking), the difference must be predominantly attributed to the intra-subject variability of the pacer. This is because the requirement of accompanied walking imposed by the experimental protocol determined the walking speed of the test subjects, hence their stride frequency and stride length, during each test. This is to say that the gait characteristics exhibited by the pacer during recording of the walking gait data for use in the avatar animation (i.e. Pacer$^{VR}$), as described in §2.1.1, and those during the tests in RL (i.e. Pacer$^{RL}$) differed. Therefore, the comparison of average gait parameters for walking SbS is not informative as to the influence of VR on gait characteristics.

Overall, it is concluded that the effect of reduction in gait parameters pertaining to the spatial and temporal reference frames, associated with VR immersion [36,62–68], is not evident in the obtained results. However, it needs to be borne in mind that the current study was not designed to address this particular problem.

## 4.2. Gait variability

The values of gait variability obtained for the pacer are slightly lower than the average gait variability for overground walking at normal speeds [98]. This is deemed acceptable since the protocol requirement of the gait of the pacer to exhibit some variability had to be reconciled with the requirement of having to produce a gait rhythm at the predefined (mean) frequency. Another acceptable circumstance is the difference between mean CoV of stride frequency of the real-life and virtual pacer at 0.42%.

The analysis of the gait variability of test subjects did not highlight any noteworthy dissimilarities between the two experimental conditions, except S5 exhibiting more variable gait in the VR environment.

## 4.3. Synchronization strength

The effect of the instruction to synchronize steps on the synchronization strength index obtained for walking in the VR environment is comparable to that obtained in the RL environment. This is the case for both considered topological arrangements (i.e. SbS and FtB). However, quantitative and qualitative differences were found between gait coordination patterns from corresponding tests in the RL and VR environments.

### 4.3.1. The effect of the instruction to synchronize steps

The mean synchronization strength indices obtained in the RL and VR environments are comparable under US for walking SbS. For walking FtB, the index obtained in the VR environment is much lower than that obtained in the RL environment. The magnitude of synchronization strength index under US is generally low, and there is a considerable spread of the mean circular direction data, suggesting the behaviour of pedestrians is highly random and any persistent gait coordination patterns are seldom present. This is unlike the results for walking under IS.

The mean synchronization strength indices under IS are much higher than those obtained under US and so is the variability of the results. This is due to the conscious effort spent on gait coordination, as dictated by the experimental protocol, and considerable intra- and inter-subject variability in this respect. The synchronization strength index for walking FtB is consistently higher than that for walking SbS for tests conducted in the same visual conditions. This agrees with the results of previous investigations probing the influence of pedestrians' topological arrangement on the synchronization strength for walking in the RL environment [16,19].

### 4.3.2. Comparison with previous studies on walking in dyads

The vast majority of walks performed under the US was characterized by low synchronization strength index, much below the theorized synchronization threshold of 0.21 [12]. This is in line with findings from van Ulzen et al. [9,10]. However, the current results stand in opposition to those from some other studies reporting that the gait synchronization in walking dyads is rather widespread, with synchronization signatures present in 25–70% of performed walks, as shown in table 2. This is believed to be caused

**Table 2.** The probability of occurrence of synchronization in dyadic walking after selected authors.

| author | no. pairs | mode | distance/ duration | quantification method | results | comments |
|---|---|---|---|---|---|---|
| Chambers et al. [4] | 441 | side by side, overground | up to 30 s | estimation of ankles' vertical displacement based on Youtube videos | 35–70%—auditory and visual cues | |
| Harrison & Richardson [14] | 6 | front to back, overground | 35 m | Knees' flexion and extension using electrogoniometers | 40%—auditory and visual cues | |
| Nessler & Gilliland [5] | 20 | side by side, treadmill | 69.6 ± 3.6 m* | Ankles' trajectories on the sagittal plane using MCS | 33.4 ± 30.1%—auditory and visual cues | |
| Nessler et al. [6] | 24 | tide by side, treadmill | 60 s | right lateral malleolus' trajectory using MCS | 50%—auditory and visual cues | |
| van Ulzen et al. [9] | 11 | side by side, treadmill | 593 m* | lower legs' movements using movement registration system | transient episodes in up to 36% of pairs— auditory and visual cues | variable walking speed |
| Zivotofsky & Hausdorff [11] | 14 | side by side, overground | 15 m | visual assessment of lower extremities' motion by two physical therapists | 29%—auditory only; 7%—visual only | obstructed cues |
| Zivotofsky et al. [12] | 14 | side by side, overground | 70 m | vertical accelerations of lower backs using accelerometers attach at L5 | 36%—auditory only; 14%—visual only | obstructed cues |
| Zivotofsky et al. [13] | 16 | side by side, overground | 70 m | vertical accelerations of lower backs using accelerometers attach at L5 | 25%—auditory and visual cues | |

*Calculated based on the test duration and walking speed values given by the original authors.

by insufficiently long walking paths adopted in those studies, ranging from 15 to 70 m. In comparison, each test subject covered the distance of approximately 286 m along the entire walking path (approx. 150 m of straight sections) during a single walk in this study, and 593 m in the study reported by van Ulzen et al. [9], albeit conducted for walking on a treadmill rather than overground. The problem with the quantification of synchronization strength based on the phase difference distribution, using signals of short duration, is illustrated here based on a theoretical argument and the data collected in this study.

The evidence suggests [9,10,18] that the coordination of gait between a pair of walkers is most often not a persistent occurrence, which can be maintained throughout an extended period. Instead, gait synchronization emerges spontaneously, and it dies out after a few gait cycles. This transient nature of the phenomenon, together with a relatively narrow range of pedestrian stride frequencies adopted in walking gait, can be the root cause of a systematic error in the quantification of synchronization strength index.

If the analysed gait signals are too short, the most likely outcome is the apparent high values of synchronization strength index. This is because the evolution of phase difference in time for two pedestrians walking with similar, but not the same, stride frequencies is relatively slow. For example, imagine two pedestrians walking at the same speed next to each other, hence their walk having the dyadic property, but adopting different, time-invariant stride frequencies: 0.96 and 0.98 Hz. Therefore, the expected synchronization strength index is zero. Since the synchronization strength index relies on a histogram of phase difference, this is only true if the distribution is uniform. Assuming sufficiently high sampling rate of the recorded (digital) signals, this condition is met if the duration of the analysed record is 50 s or its integer multiples. The error in synchronization strength estimates due to the deviation from this condition will generally diminish with the signal length. However, the effect of the insufficient test duration will be particularly debilitating in the case of tests lasting less than 50 s. Assuming the pedestrians' walking speed is close to that typically observed for normal walking, e.g. 1.5 m s$^{-1}$, this corresponds to the distance of 75 m.

Now, consider the synchronization strength indices calculated separately for each 10 m long straight section of the path used in this study, corresponding to approximately seven full gait cycles. The synchronization strength index under US is then $0.279 \pm 0.251$ and $0.289 \pm 0.215$ for walking in the RL and VR environments, respectively, which is above the proposed synchronization threshold [12]. This would lead to a false conclusion that the gait synchronization was ubiquitously present during US tests. On the other hand, under the IS, where the gait coordination was genuinely present, the analysis of individual straights would yield the mean synchronization strength indices higher by 20% and 30% for walking in the RL and VR environments, respectively, from those obtained by aggregating phase difference data from all straights prior to further processing.

It is concluded, based on these arguments, that the coordination of gait in overground walking in dyads with a real or virtual pacer, in the absence of sensory cues promoting synchronization other than visual and auditory, is relatively weak under US.

### 4.3.3. Dyad versus group walking

The synchronization strength indices obtained for dyad walking under IS are approximately 50% higher than those previously reported from a group of walkers [19]. This can be mainly attributed to the attention tuning. The dominant focus of attention during dyad walking in an experimental environment characterized by low-complexity, which is the case for most laboratory-based investigations, is the fellow pedestrian. The visual and auditory cues provided by the fellow pedestrian constitute relatively strong stimulus for gait coordination. By contrast, concurrent sensory stimuli from multiple pedestrians are available during walking in a group, with the most prominent gait coordination cues generated by the pedestrians in the immediate vicinity of the observer [19]. The abundance of sensory information, in this case, tends to break any persistent gait coordination patterns.

### 4.3.4. Qualitative differences between IS tests in RL and VR

Although the difference in the magnitude of mean synchronization strength indices for corresponding tests in RL and VR environments under IS is small (figure 4), it is accompanied by a distinct qualitative change in the pedestrian behaviour. On the one hand, approximately 92% of the walks performed in the RL environment are characterized by negative values of the mean circular direction. According to the sign convention adopted in this study, this means that the test subjects were most often lagging behind the pacer. On the other hand, approximately 54% of the corresponding walks in

the VR environment are characterized by positive values of the mean circular direction, indicating the test subjects were, slightly more often, leading the pacer.

Given the unidirectional nature of the interaction, i.e. the Pacer$^{VR}$ not reacting to any actions in its surroundings, this could be a consequence of the test subjects' anticipatory behaviour. It was previously speculated that such behaviour could emerge as a result of the collision avoidance mechanism present in walking FtB [99]. In order to enable more efficient gait corrections, the follower would try to arrive at the double stance phase of gait faster than the pedestrian positioned in front. This, however, does not explain the presence of anticipatory gait behaviour for walking SbS.

It was observed during some of the SbS tests in the VR, although not measured directly, that the test subject either tried to fall back slightly behind the pacer, so that the pacer was within their sight, or turned their head towards the pacer. This intermittent behaviour might have been the result of the limited horizontal field of view (FoV) provided by the HMD, confined to around 90 degrees [100]. In comparison, the unobstructed FoV of an average human spans approximately 200 degrees in the horizontal direction [101], which is significantly more than that of the HMD. It was previously shown that limiting horizontal FoV can impact one's perception of space [102], which might have led to more conscious stepping behaviour.

Another possible explanation of the anticipatory behaviour during VR walks is associated with the repetitiveness of Pacer$^{VR}$'s motions which allowed for greater predictability of its gait cycle. The effect could have been magnified by more secluded conditions offered by the virtual environment. Although the VR environment was designed to be a near-identical copy of the real space where the experimental campaign took place, VR isolated test subjects from peripheral stimuli (i.e. the research team supervising the experimental procedure) present in the RL environment. The lack of peripheral stimuli decreased test subjects' cognitive load and allowed for greater focus on Pacer$^{VR}$'s movements. However, this was not reflected by an increase of the synchronization strength index in VR environment, as the index took lower values than in RL environment.

### 4.3.5. Limitations and future work

The main limitations of the VR technology used in this study are associated with the relatively narrow FoV provided by the current generation of consumer-grade HMD, test subjects lacking the self-embodiment, i.e. virtual body presence, and unidirectional gait adaptation. Whether or not the lack of body presence can significantly affect gait in virtual environments remains unclear [35,103]. However, it is believed to contribute to the discrepancy in the performance of tasks requiring body coordination, such as those considered herein. The interaction with the pacer was unidirectional in the current study, whereas in the real world the interaction for walking in dyads would probably exhibit signs of bidirectionality.

The limited extent of visual field can be addressed by using a new generation of HMDs with a high refresh rate and wide FoV in vertical and horizontal directions. The self-embodiment can be provided by employing a greater number of motion capture cameras to minimize the influence of tracking markers' occlusions on the tracking quality, hence body reconstruction, or by fusing data from other instrumentation systems. Another layer of complexity can be added to the AI-driven virtual avatar by implementing bidirectional gait coordination with the neighbouring real walker. This would, however, require a deeper understanding of the bidirectional interactions while walking in dyads. Finally, the simulations can be extended to include a crowd of intelligent virtual walkers interacting with the VR-immersed real walker. Work is currently underway to realize this ambition.

## 5. Conclusion

The spontaneous gait coordination between a pair of pedestrians, in the absence of sensory cues promoting synchronization other than visual and auditory, is generally weak. The mean synchronization strength index derived from Shannon entropy reaches up to 0.07. This is consistent regardless of the type of visual environment and pedestrians' topological arrangement applied during the tests. Any gait synchronization patterns, if present, are in this case of transient nature and the mode of synchronization shows high variability, consistent between all walking conditions.

The instruction to synchronize gait significantly increases the synchronization strength index regardless of the type of visual environment and pedestrians' topological arrangement applied during the tests. However, quantitative and qualitative differences are found between results for walking in

the real-life and virtual environment. The mean synchronization strength index is generally higher for walking with a real rather than virtual pacer, reaching 0.56 and 0.62 for walking side-by-side and front-to-back, respectively. The corresponding values for walking in the virtual environment are 0.49 and 0.51. Walking in front-to-back arrangement consistently yields higher synchronization strength index than walking side-by-side for both visual environments. Walking in the virtual environment is characterized by higher variability of the synchronization strength index than walking in the real-life environment. The test subjects lag the pacer in close to 92% of the cases while walking in the real-life environment, but they lead the pacer in just over 54% of the cases while walking in the virtual environment. Although this may seem like a significant qualitative difference, the quantitative difference in mean circular direction is relatively small. The average values in the real-life environment are $-0.07\pi$ rad both for walking side-by-side and front-to-back. The corresponding values in the virtual environment are $-0.02\pi$ and $0.06\pi$ rad. The main difference lies in the variability of the mean circular direction, which is 3.6 and 4.8 times higher for walking side-by-side and front-to-back, respectively, in the virtual reality when quantified in terms of the standard deviation. The observed effect is probably due to the lack of self-embodiment in the virtual environment and limited field of view of the head-mounted display.

Overall, the results presented herein support the notion of the VR technology showing high promise for human locomotion studies, in the context of interpersonal gait coordination. Work is currently underway to extend the investigations on locomotion in virtual reality from walking in dyads to groups and crowds of pedestrians represented as intelligent avatars.

Ethics. The study was approved by the University of Leicester Ethics of Research Committee.

Data accessibility. The dataset supporting this study is available online at the Dryad Digital Repository: https://doi.org/10.5061/dryad.vx0k6djnr [104].

Authors' contributions. MB conceived the study. AAS-S designed the VR environment and analysed the data. Both authors were involved in developing the experimental protocol, data collection and writing the manuscript.

Competing interests. We declare we have no competing interests.

Funding. AAS-S was supported by University of Leicester's College of Science and Engineering Doctoral Studentship.

Acknowledgements. The authors acknowledge Mr Allan Rankin, Mr Harry Piercy and Target3D Ltd for providing the optical system used in the study and the generous hands-on assistance during the experiments; Mr Maksat Kalybek for support in the run-up and in the aftermath of the experimental campaign and Dr Niamh Hynes for assistance in the data collection.

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
