## [Reviewer comments · Royal Society Open Science]

Review History

RSOS-200622.R0 (Original submission)

Review form: Reviewer 1

Is the manuscript scientifically sound in its present form?

No

Are the interpretations and conclusions justified by the results?

No

Is the language acceptable?

Yes

Do you have any ethical concerns with this paper?

No

Have you any concerns about statistical analyses in this paper?

No

Recommendation?

Reject

Comments to the Author(s)

The hypothesis being tested is unclear. Was it that humans synchronise their gait when walking, or was it that an avatar in virtual reality provides the same stimulus as a human partner? Neither has been demonstrated (or disproved) in this study but the conclusion seems to claim that the latter has been shown. In any case, wouldn't haptic feedback be an important stimulus, as with the Millennium Bridge?

Review form: Reviewer 2**Is the manuscript scientifically sound in its present form?**

Yes

Are the interpretations and conclusions justified by the results?

Yes

Is the language acceptable?

Yes

Do you have any ethical concerns with this paper?

No

Have you any concerns about statistical analyses in this paper?

No

Recommendation?

Major revision is needed (please make suggestions in comments)

Comments to the Author(s)

Overall I think this paper is close to acceptable, but it has a key oversight. The authors have surveyed the virtual reality literature, but have overlooked a key part which is that participants in immersive systems tend to walk short. This is a very reliable effect and is now well studied. It is discussed in the book [34], but the seminal work is by Vicki Interrante's team, e.g.

Victoria Interrante, Brian Ries, Jason Lindquist and Lee Anderson (2007) Elucidating Factors that can Facilitate Veridical Spatial Perception in Immersive Virtual Environments, IEEE Virtual Reality, pp. 11-17

The first point is that the authors should really discuss their results in the context of underestimation of walking duration. The second is from that literature we know that key features of the VR impact walking behaviour. These include the visual fidelity of the model, in particular the presence of high frequency texture detail; the authors note two related points on the lack of a self-embodiment and field of view of the HMD, but the form of the visual flow even if restricted is important. I don't think this invalidates the results, but, for example, the authors could make the 3D model available to allow future critique and comparison. They could report the texture size for the walls and floors and thus the apparent resolution in pixels per degree. The second key point is that locomotion depends critically on latency. With what appears to be external only tracking that isn't fused with local acceleometry, the system will have a relatively high latency. I would strongly suggest reporting the end to end latency of this system, again because future models will need to take this into account.

I personally think that the self embodiment will have a dramatic impact, but this doesn't invalidate the findings. I wouldn't just dismiss this as a limitation, but flag the results (from the paper above) that indicate how the embodiment might change distance estimation.

Minor points.

I found the abstract a bit vague and not motivating the study precisely. the paper is not really about VR, so why lead off with that? I suggest starting with the problem in locomotion studies, and then the opportunity of validating in other areas.

Virtual reality is not an "emerging technology". It has become cheaper over the past few years.

"However, ecological validity of virtual reality interfaces is currently uncertain," This statement is a massive overstatement without a domain scope. Ecological validity is well established in many areas. For example

<https://onlinelibrary.wiley.com/doi/abs/10.1111/bjop.12290>

you anyway cite lots of work that has used virtual reality to study locomotion (e.g. from Bulthoff's group). I would pin the motivation on the specific difficulty of investigating walking synchronisation.

The abstract then finishes with a critique of VR, but doesn't answer the leading question about validity. Based on this, I would suggest that the title could be clearer about the main findings.

I would strike the very first "The" in the introduction.

The second paragraph gets at the point in a roundabout way: I would suggest first highlighting the difficulty of reproducible experiments and then discuss the ecological validity of alternate method.

The main commenton 2.1.1 is that very good walk simulators, and motion capture libraries exist, so why did you make your own? I would suggest making the motion files available so that, for example the quality of motion switching can be checked by future authors.

[34] is a book covering many different topics related to virtual locomotion interfaces so it might be highlighted as a general resource for this area, or chapters explicitly referenced.

I didn't see a discussion of Figure 2c, 2d; it wasn't clear why the two sensors should drift so fast out of phase unless the sampling rate was mis-configured.

I would like to see the hypotheses framed up front, especially that I eventually came to this; "This is in line with findings from van Ulzen [8,9].", I wanted to know what I should have expected at least in the real condition a priori.

"unified to velocity" -> "fused into a single velocity"

Decision letter (RSOS-200622.R0)

Dear Mr Soczawa-Stronczyk,

The editors assigned to your paper ("Gait coordination in overground walking with a virtual reality avatar") have now received comments from reviewers. We would like you to revise your paper in accordance with the referee and Associate Editor suggestions which can be found below (not including confidential reports to the Editor). Please note this decision does not guarantee eventual acceptance.

Please submit a copy of your revised paper before 19-Jun-2020. Please note that the revision deadline will expire at 00.00am on this date. If we do not hear from you within this time then it will be assumed that the paper has been withdrawn. In exceptional circumstances, extensions may be possible if agreed with the Editorial Office in advance. We do not allow multiple rounds of revision so we urge you to make every effort to fully address all of the comments at this stage. If deemed necessary by the Editors, your manuscript will be sent back to one or more of the original reviewers for assessment. If the original reviewers are not available, we may invite new reviewers.

- Data accessibility

<http://datadryad.org/submit?journalID=RSOS&manu=RSOS-200622>

- Competing interests

- Authors' contributions

- Acknowledgements

- Funding statement

Best regards,

on behalf of Professor Jon Crowcroft (Associate Editor) and Marta Kwiatkowska (Subject Editor)
openscience@royalsociety.org

Associate Editor's comments (Professor Jon Crowcroft):

Please clarify the hypothesis the paper is tackling and then look at the detailed feedback on related work.

Reviewers' Comments to Author:

Reviewer: 1

Comments to the Author(s)

The hypothesis being tested is unclear. Was it that humans synchronise their gait when walking, or was it that an avatar in virtual reality provides the same stimulus as a human partner? Neither has been demonstrated (or disproved) in this study but the conclusion seems to claim that the latter has been shown. In any case, wouldn't haptic feedback be an important stimulus, as with the Millennium Bridge?

Reviewer: 2

Comments to the Author(s)

Overall I think this paper is close to acceptable, but it has a key oversight. The authors have surveyed the virtual reality literature, but have overlooked a key part which is that participants in immersive systems tend to walk short. This is a very reliable effect and is now well studied. It is discussed in the book [34], but the seminal work is by Vicki Interrante's team, e.g.

Victoria Interrante, Brian Ries, Jason Lindquist and Lee Anderson (2007) *Elucidating Factors that can Facilitate Veridical Spatial Perception in Immersive Virtual Environments*, IEEE Virtual Reality, pp. 11-17

The first point is that the authors should really discuss their results in the context of underestimation of walking duration. The second is from that literature we know that key features of the VR impact walking behaviour. These include the visual fidelity of the model, in particular the presence of high frequency texture detail; the authors note two related points on the lack of a self-embodiment and field of view of the HMD, but the form of the visual flow even if restricted is important. I don't think this invalidates the results, but, for example, the authors could make the 3D model available to allow future critique and comparison. They could report the texture size for the walls and floors and thus the apparent resolution in pixels per degree.

The second key point is that locomotion depends critically on latency. With what appears to be external only tracking that isn't fused with local acceleometry, the system will have a relatively high latency. I would strongly suggest reporting the end to end latency of this system, again because future models will need to take this into account.

I personally think that the self embodiment will have a dramatic impact, but this doesn't invalidate the findings. I wouldn't just dismiss this as a limitation, but flag the results (from the paper above) that indicate how the embodiment might change distance estimation.

Minor points.

I found the abstract a bit vague and not motivating the study precisely. the paper is not really about VR, so why lead off with that? I suggest starting with the problem in locomotion studies, and then the opportunity of validating in other areas.

Virtual reality is not an "emerging technology". It has become cheaper over the past few years.

"However, ecological validity of virtual reality interfaces is currently uncertain," This statement is a massive overstatement without a domain scope. Ecological validity is well established in many areas. For example

<https://onlinelibrary.wiley.com/doi/abs/10.1111/bjop.12290>

you anyway cite lots of work that has used virtual reality to study locomotion (e.g. from Bulthoff's group). I would pin the motivation on the specific difficulty of investigating walking synchronisation.

The abstract then finishes with a critique of VR, but doesn't answer the leading question about validity. Based on this, I would suggest that the title could be clearer about the main findings.

I would strike the very first "The" in the introduction.

The second paragraph gets at the point in a roundabout way: I would suggest first highlighting the difficulty of reproducible experiments and then discuss the ecological validity of alternate method.

The main comment on 2.1.1 is that very good walk simulators, and motion capture libraries exist, so why did you make your own? I would suggest making the motion files available so that, for example the quality of motion switching can be checked by future authors.

[34] is a book covering many different topics related to virtual locomotion interfaces so it might be highlighted as a general resource for this area, or chapters explicitly referenced.

I didn't see a discussion of Figure 2c, 2d; it wasn't clear why the two sensors should drift so fast out of phase unless the sampling rate was mis-configured.

I would like to see the hypotheses framed up front, especially that I eventually came to this; "This is in line with findings from van Ulzen [8,9].", I wanted to know what I should have expected at least in the real condition a priori.

"unified to velocity" -> "fused into a single velocity"

Author's Response to Decision Letter for (RSOS-200622.R0)

See Appendix A.

Decision letter (RSOS-200622.R1)

Dear Mr Soczawa-Stronczyk,

It is a pleasure to accept your manuscript entitled "Gait coordination in overground walking with a virtual reality avatar" in its current form for publication in Royal Society Open Science.

Kind regards,
Andrew Dunn
Royal Society Open Science Editorial Office

on behalf of Professor Jon Crowcroft (Associate Editor) and Marta Kwiatkowska (Subject Editor)
openscience@royalsociety.org

Associate Editor Comments to Author (Professor Jon Crowcroft):

Thank you very much for your careful revision, together with clear response to the reviewers. I'm very happy with this paper now!

Appendix A

Corresponding author:

Mr Artur Soczawa-Stronczyk
School of Engineering
University of Leicester
University Road
Leicester, LE1 7RH, UK
e: aamss1@leicester.ac.uk

12th of June 2020

To:
Professor Jon Crowcroft
Associate Editor
Royal Society Open Science

Dear Professor Crowcroft,

We are sending herewith a copy of our manuscript for publication in Royal Society Open Science entitled: 'Gait coordination in overground walking with a virtual reality avatar'. We hereby certify that this manuscript consists of original, unpublished work which is not under consideration for publication elsewhere and there are no copyright or material sharing restrictions.

We have received insightful comments on the manuscript which the reviewers would like us to address. We have carefully considered these comments and made amendments to the manuscript, where necessary, which are outlined in detail in the remainder of this letter.

To facilitate the reviewing process, these changes are marked in blue in the re-submitted paper.

Thank you for your consideration of our paper. We look forward to hearing back from you.

Sincerely, in the name of both authors,

Mr Artur Soczawa-Stronczyk

Associate Editor's comments (Professor Jon Crowcroft):

Please clarify the hypothesis the paper is tackling and then look at the detailed feedback on related work.

We have added information on the hypotheses being tests, as detailed in our response to Reviewer 1 below. We have also significantly extended the discussion throughout the paper, including that referring to the related work on the subject, as detailed in our response to Reviewer 2.

Reviewer: 1

Comments to the Author(s)

The hypothesis being tested is unclear. Was it that humans synchronise their gait when walking, or was it that an avatar in virtual reality provides the same stimulus as a human partner? Neither has been demonstrated (or disproved) in this study but the conclusion seems to claim that the latter has been shown. In any case, wouldn't haptic feedback be an important stimulus, as with the Millennium Bridge?

We have re-stated the hypotheses at the end of the introductory section to make them more transparent, pointing out the sources informing and motivating our choices, as suggested by Reviewer 2:

It was hypothesised that a VR locomotor interface enabling unconstrained overground walking can evoke gait coordination patterns between a real pedestrian and a virtual reality avatar similar to those observed when walking in the corresponding conditions in the real life environment. Informed by the previous studies on walking in pairs (van Ulzen et al., 2008, 2010) and groups (Bocian et al., 2018; Pimentel et al., 2013; Ricciardelli & Pansera, 2010; Soczawa-Stronczyk et al., 2019), it was expected that the spontaneous synchronisation of gait between walkers, if present, would be of transient nature (van Ulzen et al., 2008, 2010). Given a sufficiently long walking path is provided, it was expected that the spontaneous synchronisation of gait would be relatively weak. Due to a lesser number of available synchronisation stimuli when compared to walking in a group of pedestrians (Bocian et al., 2018; Soczawa-Stronczyk et al., 2019), it was expected that the synchronisation strength in dyadic walking would be, on average, higher than that for walking in a group. Furthermore, it was expected that the instruction to synchronise steps would cause the synchronisation strength to be significantly higher than in the case of lack thereof (Soczawa-Stronczyk et al., 2019). Finally, it was expected that the synchronisation strength for walking front-to-back would be higher than that for walking side-by-side (Bocian et al., 2018; Soczawa-Stronczyk et al., 2019).

Therefore, our results, summarised in the concluding section (**Section 5**), are now carefully mapping the hypotheses stated in the introductory section. We note that Reviewer 2 is of the opinion that the '(originally submitted) paper is close to acceptable'. We hope that by making the tested hypotheses more explicit, we have addressed the reviewer's concerns.

The haptic feedback between pedestrians, e.g. achieved by holding hands, can significantly contribute to gait coordination while walking in pairs (Zivotofsky et al., 2012). However, based on our personal experience, this type of feedback is most often absent in real-world settings, as it predominantly occurs at the early stages of romantic relationships and parenthood. Consequently, this is not the problem being addressed in the current study.

The instability of the London Millennium Footbridge on its opening day is a fascinating problem still attracting considerable interest from a broad scientific community. Indeed, in that particular case, the bi-directional pedestrian-structure interaction was the origin of the destabilising forces to the structure. This interaction was mediated by the mechanical feedback between the pedestrian and the structure. However, pedestrian-induced structural instability is not the problem being addressed in the current study. A detailed discussion of this problem can be found elsewhere (e.g. Bocian et al. (2015)).

Should the reviewer wish to provide more detailed comments on our work, we would be happy to address them accordingly.

Reviewer: 2

We thank the reviewer for in-depth comments on our paper. The following paragraphs give detailed feedback on these comments and provide information on a number of changes made to the paper, which for the reviewer's convenience are marked in blue in the updated paper.

Comments to the Author(s):

Overall I think this paper is close to acceptable, but it has a key oversight. The authors have surveyed the virtual reality literature, but have overlooked a key part which is that participants in immersive systems tend to walk short. This is a very reliable effect and is now well studied. It is discussed in the book (Steinicke et al., 2013), but the seminal work is by Vicki Interrante's team, e.g. (Interrante et al., 2008). The first point is that the authors should really discuss their results in the context of underestimation of walking duration.

To inform the reader of any potential effects related to the reduction in gait parameters due to VR immersion, we have added the following paragraph to the introduction:

It is important to note that VR comes with certain inherent limitations. Previous comparative studies on gait parameters during walking in real and virtual environment found underestimation of egocentric distances (Interrante et al., 2008) as well as decrease of the walking speed and the stride length a common response to VR immersion (Mohler et al., 2007). This was suggested to be caused by the altered distance perception in virtual reality, often referred to as distance compression (Janeh et al., 2017; Loomis & Knapp, 2003; Renner et al., 2013; Steinicke et al., 2010; Willemsen et al., 2004), which may become negligible after a 5 minutes period of habituation to virtual environment (Mohler et al., 2010). Some more recent studies found walking and the motion adaptation in VR to be quantitatively comparable to those in the real world environment (Berton et al., 2019; Bühler & Lamontagne, 2018; Fink et al., 2007; Janeh et al., 2017; Olivier et al., 2018). A widely acknowledged consensus is that while quantitative differences between virtual and real-life locomotion might still exist, the pedestrian behaviour in both environments is qualitatively compatible (Agethen et al., 2018; Berton et al., 2019; Gérin-Lajoie et al., 2008; Moussaïd et al., 2016; Olivier et al., 2018).

To assess our results in the context of a reduction in gait parameters, the average stride frequency (i.e. reciprocal of the gait cycle duration) and the average stride length were calculated for each test and compared between those obtained for walking in RL and VR environment. Detailed information on these analyses is given in **Section 3.1** (see the first three paragraphs), but the most critical findings are provided here for the reviewer's convenience.

The Welch's t-test (Welch, 1947) at 5% significance level ($p = 0.05$) was used to test the hypotheses that the means of the stride frequency and stride length were equal. The data collected under US only were used in this analysis since the instruction to synchronise steps (in IS) overridden naturally occurring gait patterns, as discussed in **Section 4.3.1**. Moreover, walks in SbS and FtB were considered separately due to the difference in the distance travelled by test subjects in these two topological arrangements.

We found no statistically significant difference in the stride frequency (hence stride duration) for walking SbS and FtB. We found a statistically significant difference in the stride length for walking SbS, but not for walking FtB.

A detailed discussion of these findings is provided in **Section 4.1**:

*The average stride frequency and stride length of test subjects for walking SbS and FtB under US was similar between the corresponding tests conducted in RL and VR environment. This is apart from the average stride length in walking SbS for which there was a statistically significant difference. However, this occurrence cannot be considered as evidence supporting the notion of gait parameters being adversely affected by the VR immersion. This is because the average duration of tests in RL and VR environment was in this case different, at 206 s and 230 s, respectively. This is not the case for walking FtB, for which the average duration of tests in RL and VR environment was similar, at 231 s and 228 s, respectively. In both cases (SbS and FtB walking) the difference must be predominantly attributed to the intra-subject variability of the pacer. This is because the requirement of accompanied walking imposed by the experimental protocol determined the walking speed of the test subjects, hence their stride frequency and stride length, during each test. This is to say that the gait characteristics exhibited by the pacer during recording of the walking gait data for use in the avatar animation (i.e. Pacer^{VR}), as described in **Section 2.1.1**, and those during the tests in RL (i.e. Pacer^{RL}) differed. Therefore, the comparison of average gait parameters for walking SbS is not informative as to the influence of VR on gait characteristics.*

Overall, it is concluded that the effect of reduction in gait parameters pertaining to the spatial and temporal reference frames, associated with VR immersion (Interrante et al., 2008; Janeh et al., 2017; Loomis & Knapp, 2003; Mohler et al., 2010, 2007; Renner et al., 2013; Steinicke et al., 2010; Willemsen et al., 2004),

is not evident in the obtained results. However, it needs to be borne in mind that the current study was not designed to address this particular problem.

The second is from that literature we know that key features of the VR impact walking behaviour. These include the visual fidelity of the model, in particular the presence of high frequency texture detail; the authors note two related points on the lack of a self-embodiment and field of view of the HMD, but the form of the visual flow even if restricted is important. I don't think this invalidates the results, but, for example, the authors could make the 3D model available to allow future critique and comparison. They could report the texture size for the walls and floors and thus the apparent resolution in pixels per degree.

We agree that, in general, the fidelity of the VR model can have a significant influence on the results. During the tests conducted in this study, the visual field was to a large extent dominated by the avatar (Pacer^{VR}), that being the only animated component of which movement, including walking path and speed, determined the behaviour of the test subjects. This was a consequence of the adopted experimental protocol, enforcing accompanied walking in either front-to-back or side-by-side arrangement. The optics flow was enabled by referencing self-movement against the various components of the VR environment, including those having texture/pattern or brightness variation due to the global (Sun) light casting inner shadows.

The information on the rendering of various components of the VR environment is now included in **Section 2.5**:

*The VR environment used in the tests was created using ARCHICAD 23 software (Graphisoft SE, 2019) – a state-of-the-art building information modelling (BIM) tool. It consisted of a highly detailed representation of the Charles Wilson Sports Hall including 3D doors, windows, lighting features and basketball infrastructure. The walls and the ceiling were covered in solid colours and the floor was rendered using a dark monotone carpet tile texture with the resolution of 288 px/m. The appearance of these components was closely matching the real world environment. To increase realism, a global (Sun) light was positioned outside the modelled room. The global light was casting inner shadows, thus providing variable brightness surface patterns facilitating optic flow and distance estimation. An exemplar VR scene used during the experimental campaign is available as a part of the dataset supporting this study (see **Data Accessibility**).*

The number of pixels per (visual) degree is perhaps not the most informative here as the test subject was moving relative to the environment (rather than having their head fixed in space, e.g. by resting it on a chin-rest, as is often the case in psychophysics studies) hence the relative distance from the rendered objects changed throughout the test.

To enable critique and comparison of our results we have included a 3D model of the VR environment, together with an exemplar avatar behaviour, as a part of the dataset supporting our study (Reviewer URL).

The second key point is that locomotion depends critically on latency. With what appears to be external only tracking that isn't fused with local acceleometry, the system will have a relatively high latency. I would strongly suggest reporting the end to end latency of this system, again because future models will need to take this into account.

The end-to-end latency is the total time elapsed between the motion taking place and the changes in the image associated with that motion being displayed in the HMD. Therefore, it is the sum of latencies of: (i) MCS cameras, (ii) MCS data processing (MCS), (iii) data transmission (from MCS to the backpack PC), (iv) virtual scene rendering, and (v) the scene display in the HMD. We have measured the latency for (i) and (ii), and (iii) was estimated based on the median value of a typical consumer-grade, uncongested wireless network (Suiy et al., 2016).

The latencies for (i), (ii) and (iii) add up to a total of approximately 8 ms. This rather low latency is a consequence of tracking two rigid bodies only, rather than a full set of markers enabling body motion to be reconstructed. Therefore, fewer solvers were required in the online processing of kinematic data. It is worth pointing out that the latency of 8 ms is the expected overhead due to using an external tracking solution, i.e. OptiTrack MCS, rather than the proprietary motion trackers of Oculus Rift. Although we have not measured the latency for (iv) and (v) directly, a nearly identical experimental setup was previously used to study gaze behaviour during collision avoidance in walking (Berton et al., 2019). Furthermore, no discomfort due to the visual information delay was reported in post hoc interviews with the test subjects. If the latency had had a strong debilitating effect on our results, we would have expected the *mean* phase difference to differ significantly between walking in RL and VR environment under the instruction to synchronise steps (IS). However, this was not the case – the values of mean circular direction were -0.07π for walking in both SbS and FtB in RL, and -0.02π and 0.06π for walking SbS and FtB in VR, respectively, as stated in **Section 3.3.3** and the concluding **Section 5**.

We have added the following information at the end of the 1st paragraph in **Section 2.5**:

*The MCS latency, defined as the time elapsed from the cameras' exposure to the tracking data packages fully solved by Motive software and ready for transmission over IEEE 802.11n-2009 wireless network, was measured ex post facto. It did not exceed 4.7 ms. The latency of the data transmission over IEEE 802.11n-2009 wireless network was estimated to be approximately 3 ms, based on the median value of a typical consumer-grade, uncongested wireless network (Suiy et al., 2016). Although the latencies due to scene rendering and display in the HMD were not directly measured, a similar setup was previously used in (Berton et al., 2019). No discomfort due to the visual information delay was reported in post hoc interviews with the test subjects. If the latency had had a strong debilitating effect on our results, we would have expected the mean phase difference to differ significantly between corresponding tests in RL and VR environment under the instruction to synchronise steps (IS). However, this was not the case, as discussed in **Section 3.3.3**.*

I personally think that the self embodiment will have a dramatic impact, but this doesn't invalidate the findings. I wouldn't just dismiss this as a limitation, but flag the results (from the paper above) that indicate how the embodiment might change distance estimation.

We agree with the reviewer in that the virtual presence (or self-embodiment) could improve gait coordination. We have stated this in **Section 4.3.5** discussing the limitations of the developed VR platform. Interestingly, it seems the evidence for body presence significantly affecting gait in VR is currently unclear (Canessa et al., 2019; Valkov et al., 2016). However, it is likely to be a significant factor in tasks requiring coordination of motor activities, such as interpersonal synchronisation in walking.

Unless we are missing something here, the paper cited by the reviewer (Interrante et al., 2008) does not address this issue. Instead, it focuses on egocentric distance estimation without the self-embodiment. Therefore, we refrained from making further changes on this point.

Minor points

I found the abstract a bit vague and not motivating the study precisely. the paper is not really about VR, so why lead off with that? I suggest starting with the problem in locomotion studies, and then the opportunity of validating in other areas. Virtual reality is not an "emerging technology". It has become cheaper over the past few years.

"However, ecological validity of virtual reality interfaces is currently uncertain," This statement is a massive overstatement without a domain scope. Ecological validity is well established in many areas. For example: <https://onlinelibrary.wiley.com/doi/abs/10.1111/bjop.12290> you anyway cite lots of work that has used virtual reality to study locomotion (e.g. from Bulthoff's group). I would pin the motivation on the specific difficulty of investigating walking synchronisation.

All of the points mentioned above raised by the reviewer refer to the abstract. Therefore, we have rewritten the first part of the abstract to emphasize and narrow down the context of our work, and to remove statements identifying VR as an emerging technology.

It is now stated that:

Little information is currently available on interpersonal gait synchronisation in overground walking. This is caused by difficulties in continuous gait monitoring over many steps while ensuring repeatability of experimental conditions. These challenges could be overcome by utilising immersive virtual reality (VR), assuming it offers ecological validity.

To this end, this study provides some of the first evidence of gait coordination patterns for overground walking dyads in VR. Six subjects covered the total distance of 27 km while walking with a pacer. The pacer was either a real human subject or their anatomically and biomechanically representative VR avatar driven by an artificial intelligence algorithm. Side-by-side and front-to-back arrangements were tested without and with the instruction to synchronise steps.

Little evidence of spontaneous gait coordination was found in both visual conditions, but persistent gait coordination patterns were found in the case of intentional synchronisation. Front-to-back rather than side-by-side arrangement consistently yielded in the latter case higher mean synchronisation strength index.

Although the mean magnitude of synchronisation strength index was overall comparable in both visual conditions when walking under the instruction to synchronise steps, quantitative and qualitative differences were found which might be associated with common limitations of VR solutions.

The abstract then finishes with a critique of VR, but doesn't answer the leading question about validity.

Overall, our findings give evidence supporting the suitability of the developed VR platform for studying pedestrians' stepping behaviour. The 3rd paragraph of the (amended) abstract cited above informs the reader that:

- i. in both visual conditions, the spontaneous gait synchronisation is weak;
- ii. in both visual conditions, persistent gait synchronisation patterns are found for intentional synchronisation;
- iii. in both visual conditions, front-to-back rather than side-by-side topological arrangement yields higher mean synchronisation strength index.

The 4th (last) paragraph of the abstract states that the magnitude of synchronisation strength index is, on average, also compatible in both visual conditions where gait synchronisation is the most dominant pedestrian behaviour (i.e. intentional synchronisation; IS). However, some differences between pedestrians' behaviour in the real life and VR environment still exist. This is consistent with some previous studies probing the validity of VR solutions in capturing pedestrian behaviour in real-life environment, which found the results generally compatible, within certain limits (Berton et al., 2019; Bühler & Lamontagne, 2018; Fink et al., 2007; Janeh et al., 2017; Olivier et al., 2018). Therefore, we could not think of any better way of stating the results in the abstract while avoiding any oversimplifications and adhering to the imposed word limit.

However, due to a more generous word limit, a more detailed assessment of our results follows in the concluding section. The last paragraph of that section now begins with the statement that '*Overall, the results presented herein support the notion of the VR technology showing high promise for human locomotion studies, in the context of interpersonal gait coordination*'.

Based on this, I would suggest that the title could be clearer about the main findings.

We have modified the abstract but retained the title, as we believe it reflects the content of the paper well while being succinct enough.

I would strike the very first "The" in the introduction.

Deleted.

The second paragraph gets at the point in a roundabout way: I would suggest first highlighting the difficulty of reproducible experiments and then discuss the ecological validity of alternate method.

We have rewritten the second paragraph of the introduction according to the reviewer's suggestion. It now states:

Only few studies investigated spontaneous gait synchronisation in real life environment (Bocian et al., 2018; Chambers et al., 2019; Pimentel et al., 2013; Soczawa-Stronczyk et al., 2019). The main reasons for this pertain to controllability of experimental conditions and observability of measured variables. On the one hand, the desire of closely controlled setting is difficult to realise in an environment subjected to various disturbances. On the other hand, the inference of spatial and temporal gait variables demands a distributed instrumentation system capable of simultaneous capture of data generated by multiple pedestrians. Consequently, alternative methods of investigating gait adaptations between pedestrians were proposed, predominantly relying on treadmills. However, the ecological validity of the results from studies other than those enabling overground walking can be put in question (Alton et al., 1998; Stolze et al., 1997).

The main comment on 2.1.1 is that very good walk simulators, and motion capture libraries exist, so why did you make your own? I would suggest making the motion files available so that, for example the quality of motion switching can be checked by future authors.

We built our own motion simulator to satisfy the compatibility requirement of Pacer^{RL} and Pacer^{VR}. This is stated in the first paragraph of **Section 2.4**, introducing the experimental protocol. However, to make our motivations clearer and known to the reader upfront, we have added the following statement in the first paragraph of **Section 2.1** introducing the developed VR platform:

A bespoke experimental platform was developed for the purpose of this study, relying on an immersive VR environment and a distributed motion capture system. The VR environment contained a virtual pedestrian, of which gait characteristics could be closely controlled, within a physical space closely resembling a real world environment in which the experimental campaign took place. The main idea underlying the development of the experimental platform was to enable kinematic gait data to be recorded during dyadic walking with a pacer, that being either a real human or their anatomically- and biomechanically-representative VR avatar.

We have generated an executable file for readers to be able to experience our motion generator, which also includes auditory information. The file is available as a part of the dataset supporting our study (Reviewer URL). However, motion switching (referred to as blending in the paper) is handled procedurally by the game engine, i.e. Unity 2018.4.0f1 (Unity Technologies, 2019) based on the input from the steering system. Because the steering system, i.e. Polarith AI 1.6 (Polarith UG, 2019), is based on a commercial solution, we are contractually obliged not to distribute it in any form, including as a part of a package open to modifications. Therefore, we are not able to address this wish.

Steinicke et al. (2013) is a book covering many different topics related to virtual locomotion interfaces so it might be highlighted as a general resource for this area, or chapters explicitly referenced.

We have now made use of the reference pointed out by the reviewer by citing it as a general source of information on the subject on page 2:

(For a detailed review of developments on human locomotion in VR the reader is referred to Steinicke et al. (2013).)

I didn't see a discussion of Figure 2c, 2d; it wasn't clear why the two sensors should drift so fast out of phase unless the sampling rate was mis-configured.

Figures 2 (c) & (d) are discussed in **Section 2.6.2** (right under the 3rd equation) as they present the time evolution and the histogram of phase difference, respectively, explained in that section. However, for convenience of presentation and to show the step-by-step derivation of the phase difference distribution, subsequently used to quantify the synchronisation strength and directionality, **Figure 2** brings all steps in the signal processing into one place. This starts from the (truncated) signals captured from a test subject (acceleration) and Pacer^{VR} (displacement) in **Figure 2 (a)**, moving on to the corresponding velocity signals in **Figure 2 (b)**, moving on to the corresponding time history of phase difference in **Figure 2 (c)**, ending up with the corresponding histogram of phase difference in **Figure 2 (d)**.

We have added the following sentence right before **Figure 2**:

Figure 2 (c) & (d) is included here for the clarity of presentation, although the relevant discussion is given in **Section 2.6.2**

The rate of evolution of the phase difference represents a genuine frequency mismatch between the test subject and the Pacer^{VR} rather than the drift in sensors' clocks.

I would like to see the hypotheses framed up front, especially that I eventually came to this; "This is in line with findings from van Ulzen et al. (2008, 2010).", I wanted to know what I should have expected at least in the real condition a prior.

We have restated the hypotheses to make them clearer:

It was hypothesised that a VR locomotor interface enabling unconstrained overground walking can evoke gait coordination patterns between a real pedestrian and a virtual reality avatar similar to those observed when walking in the corresponding conditions in the real life environment. Informed by the previous studies on walking in pairs (van Ulzen et al., 2008, 2010) and groups (Bocian et al., 2018; Pimentel et al., 2013; Ricciardelli & Pansera, 2010; Soczawa-Stronczyk et al., 2019), it was expected that the spontaneous synchronisation of gait between walkers, if present, would be of transient nature (van Ulzen et al., 2008, 2010). Given a sufficiently long walking path is provided, it was expected that the spontaneous synchronisation of gait would be relatively weak. Due to a lesser number of available synchronisation stimuli when compared to walking in a group of pedestrians (Bocian et al., 2018; Soczawa-Stronczyk et al., 2019), it was expected that the synchronisation strength in dyadic walking would be, on average, higher than that for walking in a group. Furthermore, it was expected that the instruction to synchronise steps would cause the synchronisation strength to be significantly higher than in the case of lack thereof (Soczawa-Stronczyk

et al., 2019). Finally, it was expected that the synchronisation strength for walking front-to-back would be higher than that for walking side-by-side (Bocian et al., 2018; Soczawa-Stronczyk et al., 2019).

Therefore, our results, summarised in the concluding section, are now closely mapping the hypotheses stated in the introductory section.

"unified to velocity" -> "fused into a single velocity"

We have reworded the relevant sentence in **Section 2.6.1**:

Next, the AHRS signals (acceleration in m/s^2) and the up-sampled signals from the game engine (displacement in m) were brought to a common kinematic variable, that being velocity (expressed in m/s).

References

- Agethen, P., Sekar, V. S., Gaisbauer, F., Pfeiffer, T., Otto, M., & Rukzio, E. (2018). Behavior analysis of human locomotion in the realworld and virtual reality for the manufacturing industry. *ACM Transactions on Applied Perception*, 15(3). <https://doi.org/10.1145/3230648>
- Alton, F., Baldey, L., Caplan, S., & Morrissey, M. C. (1998). A kinematic comparison of overground and treadmill walking. *Clinical Biomechanics*, 13(6), 434–440. [https://doi.org/10.1016/S0268-0033\(98\)00012-6](https://doi.org/10.1016/S0268-0033(98)00012-6)
- Berton, F., Olivier, A. H., Bruneau, J., Hoyet, L., & Pettre, J. (2019). Studying gaze behaviour during collision avoidance with a virtual walker: Influence of the virtual reality setup. *26th IEEE Conference on Virtual Reality and 3D User Interfaces, VR 2019 - Proceedings*, 717–725. <https://doi.org/10.1109/VR.2019.8798204>
- Bocian, M., Brownjohn, J. M. W., Racic, V., Hester, D., Quattrone, A., Gilbert, L., & Beasley, R. (2018). Time-dependent spectral analysis of interactions within groups of walking pedestrians and vertical structural motion using wavelets. *Mechanical Systems and Signal Processing*, 105, 502–523. <https://doi.org/10.1016/j.ymsp.2017.12.020>
- Bocian, M., Macdonald, J. H. G., Burn, J. F., & Redmill, D. (2015). Experimental identification of the behaviour of and lateral forces from freely-walking pedestrians on laterally oscillating structures in a virtual reality environment. *Engineering Structures*, 105, 62–76. <https://doi.org/10.1016/j.engstruct.2015.09.043>
- Bühler, M. A., & Lamontagne, A. (2018). Circumvention of Pedestrians while Walking in Virtual and Physical Environments. *IEEE Transactions on Neural Systems and Rehabilitation Engineering*, 26(9), 1813–1822. <https://doi.org/10.1109/TNSRE.2018.2865907>
- Canessa, A., Casu, P., Solari, F., & Chessa, M. (2019). Comparing real walking in immersive virtual reality and in physical world using gait analysis. *VISIGRAPP 2019 - Proceedings of the 14th International Joint Conference on Computer Vision, Imaging and Computer Graphics Theory and Applications*, 2, 121–128. <https://doi.org/10.5220/0007380901210128>
- Chambers, C., Kong, G., Wei, K., & Kording, K. (2019). Pose estimates from online videos show that side-by-side walkers synchronize movement under naturalistic conditions. *PLoS ONE*, 14(6), e0217861. <https://doi.org/10.1371/journal.pone.0217861>
- Fink, P. W., Foo, P. S., & Warren, W. H. (2007). Obstacle Avoidance During Walking in Real and Virtual Environments. *ACM Transactions on Applied Perception*, 4(1), 2. <https://doi.org/10.1145/1227134.1227136>
- Gérin-Lajoie, M., Richards, C. L., Fung, J., & McFadyen, B. J. (2008). Characteristics of personal space during obstacle circumvention in physical and virtual environments. *Gait and Posture*, 27(2), 239–247. <https://doi.org/10.1016/j.gaitpost.2007.03.015>
- Graphisoft SE. (2019). *ARCHICAD* (No. 23). <https://www.graphisoft.com/archicad/>
- Interrante, V., Ries, B., Lindquist, J., Kaeding, M., & Anderson, L. (2008). Elucidating factors that can facilitate veridical spatial perception in immersive virtual environments. *Presence: Teleoperators and Virtual Environments*. <https://doi.org/10.1162/pres.17.2.176>
- Janeh, O., Langbehn, E., Steinicke, F., Bruder, G., Gulberti, A., & Poetter-Nerger, M. (2017). Walking in virtual reality: Effects of manipulated visual self-motion on walking biomechanics. *ACM Transactions on Applied Perception*, 14(2). <https://doi.org/10.1145/3022731>
- Loomis, J., & Knapp, J. (2003). Visual Perception of Egocentric Distance in Real and Virtual Environments. In *Virtual and Adaptive Environments* (pp. 21–46). CRC Press. <https://doi.org/10.1201/9781410608888.pt1>
- Mohler, B. J., Campos, J. L., Weyel, M. B., & Bühlhoff, H. H. (2007). Gait parameters while walking in a head-mounted display virtual environment and the real world. *Proceedings of the 13th Eurographics Symposium on Virtual Environments*, 85–88. <https://doi.org/10.2312/PE/VE2007Short/085-088>

- Mohler, B. J., Creem-Regehr, S. H., Thompson, W. B., & Bühlhoff, H. H. (2010). The effect of viewing a self-avatar on distance judgments in an HMD-based virtual environment. *Presence: Teleoperators and Virtual Environments*, 19(3), 230–242. <https://doi.org/10.1162/pres.19.3.230>
- Moussaïd, M., Kapadia, M., Thrash, T., Sumner, R. W., Gross, M., Helbing, D., & Hölscher, C. (2016). Crowd behaviour during high-stress evacuations in an immersive virtual environment. *Journal of the Royal Society Interface*, 13(122), 20160414. <https://doi.org/10.1098/rsif.2016.0414>
- Olivier, A. H., Bruneau, J., Kulpa, R., & Pettre, J. (2018). Walking with Virtual People: Evaluation of Locomotion Interfaces in Dynamic Environments. *IEEE Transactions on Visualization and Computer Graphics*, 24(7), 2251–2263. <https://doi.org/10.1109/TVCG.2017.2714665>
- Pimentel, R. L., Araújo, M. C., Brito, H. M. B. F., & de Brito, J. L. V. (2013). Synchronization among Pedestrians in Footbridges Due to Crowd Density. *Journal of Bridge Engineering*, 18(5), 400–408. [https://doi.org/10.1061/\(ASCE\)BE.1943-5592.0000347](https://doi.org/10.1061/(ASCE)BE.1943-5592.0000347)
- Polarith UG. (2019). *Polarith AI* (1.6). <https://polarith.com/ai/>
- Renner, R. S., Velichkovsky, B. M., & Helmert, J. R. (2013). The Perception of Egocentric Distances in Virtual Environments - A Review. *ACM Comput. Surv.*, 46(2). <https://doi.org/10.1145/2543581.2543590>
- Ricciardelli, F., & Pansera, A. (2010). An experimental investigation into the interaction among walkers in groups and crowds. *Proceedings of the 10th International Conference on Recent Advances in Structural Dynamics*.
- Soczawa-Stronczyk, A. A., Bocian, M., Wdowicka, H., & Malin, J. (2019). Topological assessment of gait synchronisation in overground walking groups. *Human Movement Science*, 66, 541–553. <https://doi.org/10.1016/j.humov.2019.06.007>
- Steinicke, F., Bruder, G., Jerald, J., Frenz, H., & Lappe, M. (2010). Estimation of detection thresholds for redirected walking techniques. *IEEE Transactions on Visualization and Computer Graphics*. <https://doi.org/10.1109/TVCG.2009.62>
- Steinicke, F., Visell, Y., Campos, J. L., & Lécuyer, A. (2013). Human walking in virtual environments: Perception, technology, and applications. In *Human Walking in Virtual Environments: Perception, Technology, and Applications* (Vol. 9781441984). Springer New York. <https://doi.org/10.1007/978-1-4419-8432-6>
- Stolze, H., Kuhtz-Buschbeck, J. P., Mondwurf, C., Boczek-Funcke, A., Jöhnk, K., Deuschl, G., & Illert, M. (1997). Gait analysis during treadmill and overground locomotion in children and adults. *Electroencephalography and Clinical Neurophysiology - Electromyography and Motor Control*, 105(6), 490–497. [https://doi.org/10.1016/S0924-980X\(97\)00055-6](https://doi.org/10.1016/S0924-980X(97)00055-6)
- Suiy, K., Zhou, M., Liu, D., Ma, M., Pei, D., Zhao, Y., Li, Z., & Moscibroda, T. (2016). Characterizing and improving WiFi latency in large-scale operational networks. *MobiSys 2016 - Proceedings of the 14th Annual International Conference on Mobile Systems, Applications, and Services*, 347–360. <https://doi.org/10.1145/2906388.2906393>
- Unity Technologies. (2019). *Unity* (2018.4.0f1). <https://unity.com/>
- Valkov, D., Martens, J., & Hinrichs, K. (2016). Evaluation of the effect of a virtual avatar's representation on distance perception in immersive virtual environments. *Proceedings - IEEE Virtual Reality*. <https://doi.org/10.1109/VR.2016.7504775>
- van Ulzen, N. R., Lamoth, C. J. C., Daffertshofer, A., Semin, G. R., & Beek, P. J. (2008). Characteristics of instructed and uninstructed interpersonal coordination while walking side-by-side. *Neuroscience Letters*, 432(2), 88–93. <https://doi.org/10.1016/j.neulet.2007.11.070>
- van Ulzen, N. R., Lamoth, C. J. C., Daffertshofer, A., Semin, G. R., & Beek, P. J. (2010). Stability and variability of acoustically specified coordination patterns while walking side-by-side on a treadmill: Does the seagull effect hold? *Neuroscience Letters*, 474(2), 79–83. <https://doi.org/10.1016/j.neulet.2010.03.008>
- Welch, B. L. (1947). The generalisation of student's problems when several different population variances are involved. *Biometrika*, 34(1–2), 28–35. <https://doi.org/10.1093/biomet/34.1-2.28>
- Willemsen, P., Colton, M. B., Creem-Regehr, S. H., & Thompson, W. B. (2004). The effects of head-mounted display mechanics on distance judgments in virtual environments. *Proceedings - 1st Symposium on Applied Perception in Graphics and Visualization, APGV 2004*, 35–38. <https://doi.org/10.1145/1012551.1012558>
- Zivotofsky, A. Z., Gruendlinger, L., & Hausdorff, J. M. (2012). Modality-specific communication enabling gait synchronization during over-ground side-by-side walking. *Human Movement Science*, 31(5), 1268–1285. <https://doi.org/10.1016/j.humov.2012.01.003>